# GeoPurify: A Data-Efficient Geometric Distillation Framework for Open-Vocabulary 3D Segmentation

**Weijia Dou**[1]    **Xu Zhang**[2]    **Yi Bin**[1]*    **Jian Liu**[1]    **Bo Peng**[2]
**Guoqing Wang**[3]    **Yang Yang**[3]    **Heng Tao Shen**[1]
[1]Tongji University    [2]Tianjin University
[3]University of Electronic Science and Technology of China

**Code Repository:**    https://github.com/tj12323/GeoPurify

## Abstract

Recent attempts to transfer features from 2D Vision–Language Models (VLMs) to 3D semantic segmentation expose a persistent trade-off. Directly projecting 2D features into 3D yields noisy and fragmented predictions, whereas enforcing geometric coherence necessitates costly training pipelines and large-scale, annotated 3D data. We argue that this limitation stems from the dominant *segmentation-and-matching* paradigm, which fails to reconcile 2D semantics with 3D geometric structure. The geometric cues are not eliminated during the 2D-to-3D transfer but remain latent within the noisy and view-aggregated features. To exploit this property, we propose **GeoPurify** that applies a small Student Affinity Network to purify 2D VLM-generated 3D point features using geometric priors distilled from a 3D self-supervised teacher model. During inference, we devise a Geometry-Guided Pooling module to further denoise the point cloud and ensure the semantic and structural consistency. Benefiting from latent geometric information and the learned affinity network, GeoPurify effectively mitigates the trade-off and achieves superior data efficiency. Extensive experiments on major 3D benchmarks demonstrate that GeoPurify achieves or surpasses state-of-the-art performance while utilizing only ∼**1.5%** of the training data.

## 1 Introduction

Effective 3D scene understanding is critical for applications like autonomous driving, robotics, and augmented reality. However, progress is hindered by the conventional closed-world paradigm, which assumes a fixed set of object categories. This approach fails to scale to the diverse and complex real-world objects and is further constrained by the prohibitive cost of manual 3D annotation, a notoriously laborious process that makes creating comprehensive datasets intractable (Dai et al., 2017). To move beyond these limitations, the field is shifting toward open-vocabulary 3D understanding, which enables models to identify objects using arbitrary descriptions rather than predefined labels.

This new paradigm primarily leverages the broad semantic capacity of 2D Vision-Language Models (VLMs) (Radford et al., 2021; Jia et al., 2021; Zou et al., 2023) to extend a limited set of semantic labels to an open vocabulary. Existing approaches can be grouped into two categories: training-free and training-based. Training-free methods directly exploit 2D VLMs for segmentation, projecting multi-view 2D predictions onto 3D point clouds, and merging them to obtain final outputs. However, this often results in severe geometric inconsistencies, since 2D models lack awareness of 3D spatial structure. Training-based methods, in contrast, attempt to learn point-level mappings from 3D geometry to semantic labels, but require dense point cloud annotations, which are costly and labor-intensive to obtain. As a result, current approaches are constrained by an unfavorable trade-off: either accept noisy and fragmented outputs from training-free projection or incur the heavy data and computational burden of training-based pipelines to enforce geometric coherence.

In the 2D-to-3D transfer, the 2D models typically follow a "Segmentation and Matching" paradigm (Li et al., 2022; Ghiasi et al., 2022; Wang et al., 2024a), a brittle two-step process that

---

*Corresponding author. Email: yi.bin@hotmail.com

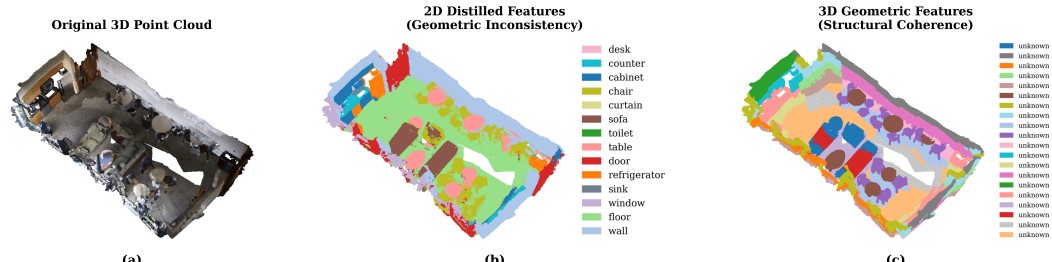

Figure 1: **The Fundamental Disconnect: Semantic Richness vs. Geometric Coherence.** *Left:* Original RGB 3D scene. *Middle:* Features distilled from 2D VLMs (Zou et al., 2023) offer rich semantics but exhibit geometric inconsistency, leading to fragmented and noisy segmentations. *Right:* Features from 3D self-supervised model (Wu et al., 2025) provide strong structural coherence with geometrically sound segments, yet inherently lack open-vocabulary semantic understanding. Our proposed method aims to bridge this critical gap by purifying the semantically rich 2D features with robust 3D geometric priors.

disconnects geometry and semantics. We argue that machine perception should instead follow a "Segmentation as Understanding" paradigm, where recognizing an object's form and understanding its meaning are intrinsically linked. However, simply adopting a generalist VLM that follows this paradigm is insufficient. As shown in Figure 1-(b), 2D VLM features ($F_{sem}$) are semantically rich but geometrically inconsistent, resulting fragments and shape distortion, whereas priors from 3D self-supervised models ($G_{geo}$) are geometrically reliable but lack open-vocabulary semantics, as illustrated in Figure 1-(c). Inspired by Wysoczańska et al. (2024), we hypothesize that transferring 2D features into 3D does not destroy the geometric information, but rather renders it latent, suggesting that we may recover latent structure more efficiently instead of learning 3D geometry from scratch.

Motivated by this hypothesis, we present **GeoPurify**, a data-efficient framework designed to recover latent geometric structure from noisy semantic features and produce robust 3D representations. We first leverage a one-for-all VLM to generate semantic-rich and geometric-aware 3D features from multiple 2D views. To address the noise and fragmentation inherent in VLM-generated point clouds, we propose Geometric Contrastive Distillation, where a student module learns latent geometric affinities from noisy features under the guidance of a 3D teacher's structural priors. By exploiting the pre-existing geometric knowledge, GeoPurify eliminates the need for large-scale manual annotation and can be trained using only a small set of unlabeled 3D scans. At inference, a Geometry-Guided Pooling module is designed to denoise and consolidate the features, producing unified 3D representations that are both semantically rich and geometrically coherent. Experiments on multiple popular 3D semantic segmentation datasets demonstrate that GeoPurify achieves comparable or superior performance to state-of-the-art methods while using only ∼**1.5%** of the training data, effectively resolving the trade-off that limits prior approaches.

In summary, our contributions are:

- We introduce **GeoPurify**, a data-efficient framework built on the hypothesis that beyond their semantic richness, VLM-projected features also embed a latent 3D geometric structure. GeoPurify demonstrates that recovering this latent structure is a more data-efficient path to open-vocabulary 3D segmentation.

- Our framework first leverages a generalist VLM to establish a rich semantic foundation via "Segmentation as Understanding", overcoming the low semantic ceiling of prior "Segmentation and Matching" approaches . It then introduces a Geometric Contrastive Distillation mechanism to learn latent geometric affinities from unlabeled 3D scans and a Geometry-Guided Pooling module that uses these affinities to denoise features and ensure structural consistency at inference.

- Extensive experiments on multiple 3D benchmarks show that GeoPurify achieves performance comparable to or better than state-of-the-art methods while using only ∼**1.5%** of the training data, highlighting its superior data efficiency. To ensure reproducibility and foster future research, we will publicly release our code and pre-trained models.

## 2 RELATED WORK

**Closed-Set 3D Scene Understanding**   Closed-set 3D scene understanding, which assumes a fixed vocabulary of object categories, is dominated by two paradigms: Voxel-based and Point-based methods (Guo et al., 2020). Voxel-based approaches discretize scenes into grids and leverage sparse convolutions for computational efficiency (Graham et al., 2018; Choy et al., 2019; Zhao et al., 2023). Point-based methods, in contrast, operate directly on raw point clouds to preserve geometric detail, evolving from the pioneering PointNet(++) (Qi et al., 2017a;b) to modern local aggregation operators (Xu et al., 2021; Wu et al., 2019) and efficient Transformer architectures (Duan et al., 2023; Liu et al., 2023; Wu et al., 2024). While highly successful on benchmarks like ScanNet (Dai et al., 2017) and S3DIS (Armeni et al., 2016), their foundational closed-set assumption prevents generalization to unseen categories, critically limiting real-world applicability and motivating the shift to open-vocabulary models.

**Open-Vocabulary 3D Scene Understanding**   To overcome closed-set limitations, open-vocabulary 3D scene understanding methods transfer rich semantics from 2D Vision-Language Models (VLMs). The core challenge is rectifying geometric inconsistencies that arise when projecting multi-view 2D features into a unified 3D space. Current solutions often tackle this with extensive supervision. Some perform simple training-free feature aggregation, but this yields noisy and fragmented 3D features (Jatavallabhula et al., 2023; Lu et al., 2023; Peng et al., 2023; Nguyen et al., 2024). While subsequent post-processing methods like PoVo (Mei et al., 2025) attempt to refine these initial features, they often fail to enforce consistency for complex or occluded objects. The dominant approach is large-scale knowledge distillation, where a 3D backbone is trained on noisy pseudo-labels from 2D projections, requiring massive datasets to implicitly learn geometric priors (Chen et al., 2023b; Peng et al., 2023; Li et al., 2024; Wang et al., 2024c; Li et al., 2025; Wang et al., 2024b; Xu et al., 2024). Others attempt direct 3D-text alignment, but typically rely on coarse, region-level text supervision that lacks the granularity for dense prediction tasks (Ding et al., 2023; Yang et al., 2024; Wang et al., 2025; Jiang et al., 2024). Recent works such as OV3D (Jiang et al., 2024) and PGOV3D (Zhang et al., 2025) advance VLM-based understanding pipelines but typically rely on large-scale data for alignment or curriculum learning, missing the efficiency of explicitly recovering latent geometric affinities. Ultimately, while techniques like AFOV (Sun et al., 2025) enhance annotation efficiency, they still share a fundamental reliance on massive datasets to implicitly resolve geometric ambiguities through improved supervision. This reveals a critical gap for a data-efficient method that explicitly enforces geometric coherence during the 2D-to-3D semantic transfer.

## 3 METHODOLOGY

### 3.1 OVERALL ARCHITECTURE

As illustrated in Figure 2, our proposed **GeoPurify** first leverages a frozen Vision-Language Model ($\Psi_{2D}$) to transfer and merge multi-view RGB images into an initial 3D feature map $F_{sem} \in \mathbb{R}^{N \times D_{sem}}$. Although semantically rich, these features lack local geometric coherence. To address this, we introduce a 3D student affinity network $\phi_S$ that learns geometric affinities from the point cloud using a self-supervised 3D geometric model to refine $F_{sem}$. The student $\phi_S$ is trained without semantic labels through knowledge distillation from a frozen 3D self-supervised teacher $\phi_T$. Specifically, $\phi_S$ learns to recover the latent geometric structure embedded in $F_{sem}$ and approximate the teacher's geometric representation by minimizing an InfoNCE contrastive loss over point triplets sampled from $\phi_T$'s embedding space. At inference, the semantic-rich and geometry-aware features are further refined by a single geometric pooling operation, which enforces local consistency through information aggregation among geometrically similar points.

### 3.2 SEMANTIC INITIALIZATION FROM A GENERALIST VLM

To obtain 3D representations enriched with semantic priors, we project RGB inputs into the 3D point space (constructed by aggregating multi-view projections, without necessitating external raw point clouds at inference) using a frozen 2D VLM and merge multiple views to capture structural cues. Prior methods often follow a Segmentation and Matching paradigm, as in LSeg (Li et al., 2022),

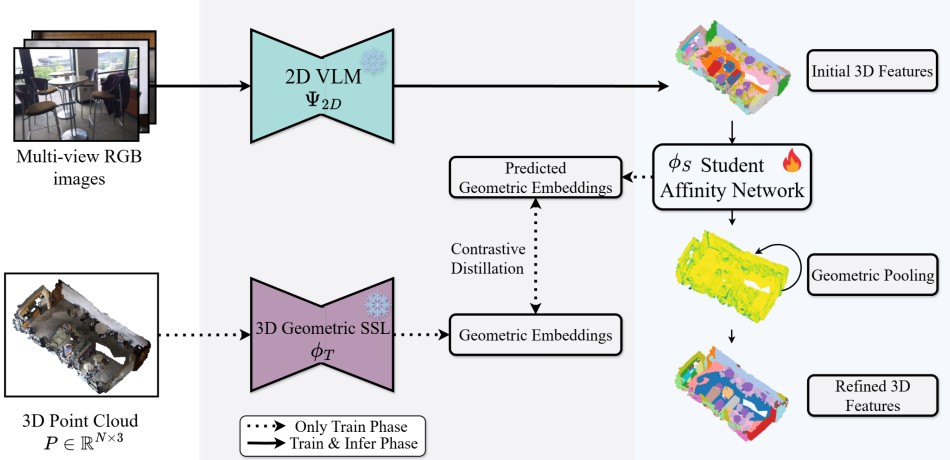

Figure 2: **GeoPurify: A Data-Efficient Pipeline for Geometric Purification of 3D Semantic Features.** Our method consists of two stages. **1) Training (left, dotted path):** A Student Affinity Network ($\phi_S$) is trained to comprehend 3D structure. It learns geometric relationships directly from the point cloud, using contrastive distillation to mimic the embeddings of a powerful, frozen 3D SSL teacher ($\phi_T$). This training phase requires no 3D semantic labels. **2) Inference (right, solid path):** A frozen 2D VLM ($\Psi_{2D}$) generates initial 3D features by projecting rich semantic content from multi-view images. These features, however, are geometrically inconsistent. The pre-trained student network then applies a geometry-aware pooling, using its learned affinities to refine the initial features. This process yields a final representation that is both semantically rich and geometrically coherent.

OpenSeg (Ghiasi et al., 2022), and SAM+CLIP frameworks (Wang et al., 2024a), which adopt a Localize then Recognize pipeline. They first group pixels into regions using segmentation backbones and then match these clusters to predefined text labels. While effective in closed-set settings, this approach produces highly discriminative yet task-specific features, limiting their semantic ceiling and reducing their capacity to capture the generative and associative richness needed in open-world 3D environments. To overcome this limitation, we adopt X-Decoder (Zou et al., 2023) as our frozen 2D VLM backbone ($\Psi_{2D}$). Instead of isolating segmentation as a downstream task, X-Decoder constructs a unified visual-linguistic embedding space that supports diverse tasks such as segmentation, retrieval, and dense captioning. Its dual-query architecture, which learns from both latent visual queries and explicit text queries, makes segmentation an emergent property of deeper contextual understanding. The resulting feature space is inherently generative and associative, optimized to answer the holistic question: *What is depicted, where is it located, and how can it be described in language?* These richer representations provide a higher semantic ceiling and yield stronger 2D priors for 3D scene understanding.

Follow this pipeline, to produce a 3D point cloud $P = \{p_i\}_{i=1}^N$ with its corresponding multi-view images $\{I_v\}_{v=1}^V$, we first compute dense feature maps for each image using $\Psi_{2D}$. For each point $p_i$, its feature $f_{i,v}$ is sampled from the feature map of each visible view $v$ via camera projection $\pi_v$. The features from multiple views are then aggregated using a weighted averaging scheme to produce a single initial semantic descriptor $f_i^{\text{sem}}$. This yields the initial feature set

$$F_{sem} = \{f_i^{\text{sem}}\}_{i=1}^N \in \mathbb{R}^{N \times D_{\text{sem}}}. \tag{1}$$

Although semantically potent, this feature aggregation from 2D views introduces geometric inconsistencies, necessitating the subsequent purification stage detailed next.

### 3.3 GEOMETRIC CONTRASTIVE DISTILLATION

To rectify the geometric inconsistencies within the initial semantic features $F_{sem}$, we introduce a contrastive purification module trained via knowledge distillation. This framework employs a

powerful, frozen 3D foundation model, Sonata (Wu et al., 2025), as the teacher ($\phi_T$) to provide a robust geometric target space, and a trainable sparse 3D CNN as the student ($\phi_S$). Crucially, the student's objective is not to replicate the teacher's features but to learn the geometric affinities between points as defined by the teacher's embedding space. The student network therefore operates directly on the point cloud geometry $P$, learning to produce a geometrically-aware embedding space that will later be used to refine $F_{sem}$.

The knowledge transfer is driven by a contrastive objective structured via an efficient hybrid sampling strategy that distills both global and local context from the teacher's feature space. For each anchor point $p_a$, we define its positive pair $p_p$ as the point with the highest feature similarity across the entire scene. We then curate a set of hard negatives comprising two distinct types: macro-negatives, which are the points globally most dissimilar to $p_a$ in the feature space, teaching the model overall scene structure; and micro-negatives, which are sampled from $p_a$'s spatial neighborhood but are specifically chosen for having the lowest feature similarity within that local region. This micro-sampling of spatially proximate yet featurally distant points forces the model to learn fine-grained geometric distinctions. The student network, $\phi_S$, which maps the point cloud $P$ to a set of geometric embeddings $G_{geo} \in \mathbb{R}^{N \times D_{geo}}$, is then optimized to organize its embedding space according to this distilled affinity information by minimizing the InfoNCE loss:

$$\mathcal{L} = -\mathbb{E}_{p_a} \left[ \log \frac{\exp(\text{sim}(g_a, g_p)/\tau)}{\exp(\text{sim}(g_a, g_p)/\tau) + \sum_{k=1}^{K} \exp(\text{sim}(g_a, g_{n_k})/\tau)} \right] \tag{2}$$

where $g \in G_{geo}$ are the geometric embeddings produced by the student $\phi_S$, $\text{sim}(\cdot, \cdot)$ is the cosine similarity, and $\tau$ is a temperature hyperparameter. We specifically choose a contrastive objective over direct feature regression, as our goal is not to replicate the teacher's feature vectors but to distill its underlying relational structure. By optimizing this objective, $\phi_S$ learns to encode the intrinsic geometric structure of the scene, producing a representation capable of enforcing structural coherence upon the initial semantic features.

### 3.4 Purifying Features via Geometry-Guided Pooling

At inference, the Geometry-Guided Pooling module refines the initial semantic features $F_{sem}$. The trained student network $\phi_S$ first processes the voxelized point cloud to generate a geometric embedding $g_i$ for each of the $V$ active voxels. From these embeddings, we construct a sparse affinity matrix $A \in \mathbb{R}^{V \times V}$ that encodes local geometric relationships. For each voxel $v_i$, we identify its $K$ nearest spatial neighbors to form a local neighborhood $\mathcal{N}(i)$. The affinity $A_{ij}$ to a neighbor $v_j \in \mathcal{N}(i)$ is computed by applying a sharpened softmax to the cosine similarity of their respective embeddings:

$$A_{ij} = \frac{\exp(\alpha \cdot \text{sim}(g_i, g_j))}{\sum_{k \in \mathcal{N}(i)} \exp(\alpha \cdot \text{sim}(g_i, g_k))} \tag{3}$$

where $\text{sim}(\cdot, \cdot)$ is the cosine similarity and $\alpha$ is a sharpening factor, analogous to an inverse temperature, that controls the concentration of the affinity weights.

The resulting matrix $A$ guides an iterative pooling process that propagates semantic information across the learned geometric graph. Initializing with the voxel-aggregated semantic features $F^{(0)}$, we update the features for $T$ steps:

$$F^{(t+1)} = AF^{(t)} \tag{4}$$

Finally, the refined voxel features $F^{(T)}$ are mapped back to their corresponding original points, yielding the purified feature set $F'_{sem}$. This iterative refinement enforces local geometric consistency by averaging features based on the learned structural affinities, effectively denoising the initial representation while preserving its semantic richness.

## 4 Experiments

### 4.1 Experimental Setup

**Datasets and Data-Efficient Training.** We evaluate our method on three prominent 3D indoor scene understanding benchmarks. Our primary evaluation is conducted on ScanNetV2 (Dai et al.,

2017), a large-scale dataset of over 1,500 RGB-D scans from diverse indoor environments, and Matterport3D (Chang et al., 2017), which contains 90 building-scale scenes with rich geometric and visual detail. For a rigorous analysis of performance on rare object categories, we also leverage the challenging long-tail benchmark ScanNet200 (Rozenberszki et al., 2022).

As aforementioned, benefiting from the geometric guidance distillation to recover the latent structure information, our GeoPurify could be learnt efficiently. We sample a highly compact subset of original training data, comprising merely **20 scenes (∼1.6%) from ScanNetV2** and **20 scene regions (∼1.3%) from Matterport3D**. Once this compact subset is selected, the subsequent distillation training is performed without using any 3D semantic labels, relying only on the raw point cloud geometry and multi-view imagery of these 20 scenes. To make the subset contain more object semantics, we leverage the ground-truth semantic statistics of the full training set to identify the subset with maximum variance. We posit that the most informative scenes are not simply those with many object types, but those where these objects appear in balanced proportions.

To formalize this, our selection process is guided by two metrics. The first, semantic richness ($N_c$), is a direct count of unique object categories. However, richness alone is insufficient, as a scene can have many categories yet be dominated by trivial structures like walls and floors. We therefore introduce a more critical metric, semantic complexity ($H_c$), measured via Shannon entropy (Shannon, 1948):

$$H_c = -\sum_{c \in C} p(c) \log_2 p(c), \tag{5}$$

where $p(c)$ is the proportion of points in category $c$. Entropy directly quantifies the scene's informational density. A high-entropy scene, such as a cluttered office, exhibits a balanced distribution of diverse objects and thus offers a rich learning signal. Conversely, a low-entropy scene, like an empty hallway, is predictable and provides limited training value.

Our selection pipeline unfolds in three stages. First, we filter for quality, culling any scene that falls below the median value for both richness ($N_c$) and complexity ($H_c$). Second, to ensure the subset spans diverse environmental archetypes (e.g., kitchens, offices), we perform K-Means clustering on the semantic category histograms of the filtered scenes. Finally, from each resulting cluster, we select the single most exemplary scene by ranking them with a composite score,

$$S = H_{c,norm} + \gamma \cdot N_{c,norm}, \tag{6}$$

which jointly rewards normalized complexity and richness. This strategy yields a training set that is not merely small but is a curated distillation of the dataset's core semantic diversity.

**Evaluation Metrics.** We evaluate our method on the task of open-vocabulary 3D semantic segmentation, focusing on the zero-shot setting. We follow the evaluation protocol established in CUA-O3D (Li et al., 2025) and report mean Intersection-over-Union (mIoU) and mean Accuracy (mAcc). Furthermore, to assess performance on a wider range of object categories from the ScanNet200 benchmark, we also report foreground mIoU (f-mIoU) and foreground mAcc (f-mAcc). Following established practice (Ding et al., 2023; Jiang et al., 2024; Yang et al., 2024; Wang et al., 2025), these foreground metrics are calculated on all classes while excluding common structural categories like "wall", "floor", and "ceiling".

**Implementation Details.** We use a pre-trained X-Decoder (Zou et al., 2023) (DaViT-L) as our frozen 2D semantic backbone ($\Psi_{2D}$) and the frozen Sonata (Wu et al., 2025) model as our geometric teacher ($\phi_T$). Our student network ($\phi_S$) is a sparse 3D network built with a series of residual blocks that produces a 128-dimensional geometric embedding for each point. The student network is trained for 50 epochs using the AdamW optimizer with a learning rate of $1 \times 10^{-3}$, which is decayed using a cosine annealing schedule. For the InfoNCE loss, the temperature $\tau$ is set to 0.07. During training, we sample 4096 anchor points per scene and, for each anchor, we sample 64 negatives (48 macro, 16 micro). At inference, the Geometry-Guided Pooling is applied for $T = 18$ iterations with an affinity sharpening factor $\alpha$ set to $1/20.0$. All experiments are conducted on a single NVIDIA L40 GPU.

## 4.2 QUANTITATIVE RESULTS

We evaluate GeoPurify against state-of-the-art methods on open-vocabulary and long-tail 3D segmentation benchmarks. The results demonstrate that by distilling a geometric prior from a small data

subset (∼1.5%), GeoPurify achieves competitive, and in some cases state-of-the-art, performance without requiring large-scale 3D semantic annotations.

**Open-Vocabulary Segmentation Performance.**  As shown in Table 1, GeoPurify demonstrates exceptional performance by leveraging a curated data subset of only ∼1.5%. To rigorously validate that our method's efficacy stems from its architecture rather than just the data selection, we re-trained a leading competitor, CUA-O3D (Li et al., 2025), on the exact same compact subset. Under these identical data-efficient conditions, CUA-O3D's performance drops sharply to an 18.1 mIoU on ScanNetV2. In stark contrast, GeoPurify achieves an mIoU of **55.1**. We attribute this robustness to our decoupled design. Standard methods typically attempt to learn entangled geo-semantic representations from scratch, failing to generalize without sufficient data. In contrast, GeoPurify relies on the frozen VLM for semantics and trains the student network solely to recover *latent* geometric structure. Learning these geometric grouping rules requires significantly less data, allowing our method to function as an efficient "geometric denoiser" that maintains high performance.

Remarkably, our model not only excels in this low-data regime but also rivals methods trained on the full dataset. The advantage is even more pronounced in the mean Accuracy (mAcc) metric, where GeoPurify attains **72.5** on ScanNetV2 and **62.4** on Matterport3D, substantially outperforming fully-trained competitors. This divergence between high mAcc and competitive mIoU is an intentional outcome of our design. The superior mAcc is driven by Geometry-Guided Pooling, which propagates VLM-initialized core labels across geometrically coherent regions, enforcing intra-object consistency and boosting recall. This mechanism, however, introduces a deliberate trade-off: by prioritizing semantic purity within objects, it can cause minor "semantic bleeding" at boundaries, slightly degrading the boundary precision to which mIoU is sensitive. For a detailed qualitative analysis, including visual examples, please refer to Appendix D.

Table 1: **Quantitative results for open-vocabulary 3D semantic segmentation on ScanNetV2 and Matterport3D.** We compare GeoPurify, trained with only ∼1.5% of the 3D data, against fully supervised (*Fully-sup.*) and zero-shot (*Zero-shot*) methods that utilize the full training set. Values in **bold** indicate the best result for each metric among the Zero-shot and Data-Efficient Zero-shot methods. [†]Denotes results from the original paper's LSeg-based implementation. [‡]Denotes results reproduced by CUA-O3D (Li et al., 2025) and MPEC (Wang et al., 2025).

| Type | Method | 3D Net Training Data | ScanNetV2 | | Matterport3D | |
|---|---|---|---|---|---|---|
| | | | mIoU | mAcc | mIoU | mAcc |
| *Fully-sup.* | TextureNet Huang et al. (2019) | 100% | 54.8 | - | - | 63.0 |
| | ScanComplete (Dai et al., 2018) | 100% | 56.6 | - | - | 44.9 |
| | DCM-Net (Schult et al., 2020) | 100% | 65.8 | - | - | 66.2 |
| | SupCon (Zheng et al., 2021) | 100% | 69.2 | 77.7 | 53.1 | 63.4 |
| | MinkowskiNet (Choy et al., 2019) | 100% | 69.2 | 77.7 | 53.1 | 63.4 |
| | LGround (Rozenberszki et al., 2022) | 100% | 73.2 | - | - | 67.2 |
| *Zero-shot* | PLA (Ding et al., 2023) | 100% | 17.7 | 33.5 | - | - |
| | CLIP2Scene (Chen et al., 2023b) | 100% | 25.1 | - | - | - |
| | CNS (Chen et al., 2023a) | 100% | 26.8 | - | - | - |
| | CLIP-FO3D (Zhang et al., 2023) | 100% | 30.2 | 49.1 | - | - |
| | RegionPLC[‡] (Yang et al., 2024) | 100% | 43.8 | 65.6 | 28.9 | 43.8 |
| | MSeg Voting (Lambert et al., 2020) | 100% | 45.6 | 54.4 | 33.4 | - |
| | DMA-text only (Li et al., 2024) | 100% | 50.5 | 63.7 | 39.8 | 49.5 |
| | DMA[†]-3D (Li et al., 2024) | 100% | 54.8 | 66.9 | - | - |
| | OpenScene[†‡]-3D (Peng et al., 2023) | 100% | 51.6 | 63.1 | 40.5 | 48.8 |
| | CUA-O3D (3D) (Li et al., 2025) | 100% | 54.1 | 64.1 | 41.3 | 49.5 |
| | GGSD (Wang et al., 2024b) | 100% | 56.5 | 68.6 | - | - |
| | OV3D (Jiang et al., 2024) | 100% | 57.3 | 72.9 | **45.8** | **62.4** |
| | PGOV3D (Zhang et al., 2025) | 100% | **59.5** | **73.2** | - | - |
| *Data-Efficient Zero-shot* | CUA-O3D[reimple] (3D) (Li et al., 2025) | ∼1.5% | 18.1 | 26.4 | 14.0 | 20.5 |
| | (*Ours*) GeoPurify | ∼1.5% | 55.1 | 72.5 | 40.2 | **62.4** |

**Generalizability on Long-Tail Datasets.**  As shown in Table 2, GeoPurify establishes a new state-of-the-art on long-tail benchmarks like ScanNet200 and the challenging M160 split. This robust generalization arises from the synergy between our semantic and geometric modules. Our generalist

VLM provides a high semantic ceiling, generating descriptive "semantic seeds" even for rare objects where traditional recognition-focused backbones might fail. These initial, often sparse, signals are then amplified by our class-agnostic geometric prior. Because this prior is distilled purely from unlabeled geometry, it is immune to the frequency bias inherent in semantic datasets. During inference, the Geometry-Guided Pooling leverages this unbiased structural knowledge to propagate the VLM's semantic seeds across entire geometrically coherent instances. In short, the VLM proposes a weak semantic hypothesis, and our geometry-aware network confidently validates and completes it, ensuring robust segmentation even for categories with scarce data.

Table 2: **Zero-shot segmentation performance on long-tail benchmarks.** We report key metrics (f-mIoU, f-mAcc) on ScanNet200 and (mIoU, mAcc) on frequency-based splits of Matterport3D (top K=40, 80, 160 classes). Our data-efficient GeoPurify is compared against other zero-shot baselines. Values in **bold** indicate the best result for each metric among methods. [†]Results from the original LSeg-based implementation. [‡]Results reproduced by DMA (Li et al., 2024) and MPEC (Wang et al., 2025).

| Method | ScanNet200 | | Matterport40 | | Matterport80 | | Matterport160 | |
|---|---|---|---|---|---|---|---|---|
| | f-mIoU | f-mAcc | mIoU | mAcc | mIoU | mAcc | mIoU | mAcc |
| PLA (Ding et al., 2023) | 1.8 | 3.1 | - | - | - | - | - | - |
| DMA-text only (Li et al., 2024) | 6.9 | 11.3 | 25.4 | 31.6 | 11.7 | 16.1 | 6.2 | 8.0 |
| DMA(FC-CLIP)-3D (Li et al., 2024) | 7.9 | 15.2 | **38.4** | **48.3** | **20.1** | 26.5 | 9.8 | 15.2 |
| OpenScene[†‡]-3D (Peng et al., 2023) | 7.3 | - | 25.4 | 30.7 | 12.0 | 15.2 | 5.9 | 7.5 |
| RegionPLC (Yang et al., 2024) | 9.1 | 17.3 | - | - | - | - | - | - |
| (*Ours*) GeoPurify | **11.9** | **22.8** | 33.1 | 48.0 | 18.6 | **31.7** | **11.9** | **20.5** |

**Cross-Dataset Generalization.** We evaluate zero-shot cross-dataset generalization to test the robustness of our learned geometric prior. As shown in Table 3, GeoPurify significantly outperforms existing methods in both transfer directions. The advantage is particularly pronounced when transferring from Matterport3D → ScanNetV2, where our method achieves **54.9** mIoU, surpassing the next-best competitor by a large margin of **16.3** points.

This striking performance gap reveals a fundamental difference in what is learned. While Matterport3D's scenes are geometrically rich, methods that learn entangled geo-semantic representations tend to overfit to this richness. Their learned priors become correlated with Matterport3D's specific object styles and semantic distribution, making them brittle when transferred to a new domain like ScanNetV2. In contrast, our decoupled training process allows GeoPurify to capitalize on the geometric complexity without semantic overfitting. By distilling knowledge based purely on structural affinities, our student network learns a **class-agnostic, geometric prior** that captures abstract shapes and relationships. This purely structural knowledge is highly transferable, acting as a robust regularizer even when the semantic context changes entirely. This result provides compelling evidence that our method learns a more fundamental and truly domain-agnostic understanding of 3D geometry. A more granular analysis of cross-domain generalization under an expanded vocabulary is presented in Appendix B.2.

Table 3: **Zero-shot cross-dataset evaluation.** Models are trained on the source dataset and evaluated directly on the target without fine-tuning.

| Method | ScanNetV2 → Matterport3D | | Matterport3D → ScanNetV2 | |
|---|---|---|---|---|
| | mIoU (%) | mAcc (%) | mIoU (%) | mAcc (%) |
| OpenScene (Peng et al., 2023) | 36.0 | 48.0 | 36.5 | 44.0 |
| CUA-O3D (Li et al., 2025) | 37.4 | 49.2 | 38.6 | 46.6 |
| GGSD (Wang et al., 2024b) | 40.1 | 54.4 | - | - |
| **GeoPurify (Ours)** | **40.5** | **62.7** | **54.9** | **71.9** |

## 4.3 ABLATION STUDIES

We conduct a series of ablation studies on the ScanNetV2 validation set to analyze the contribution of each component within our GeoPurify framework. The results, summarized in Table 4, systematically deconstruct our model to validate its core design principles.

Table 4: Ablation studies of GeoPurify on the ScanNetV2 validation set. We analyze the impact of our core geometric purification module, the choice of 2D backbone, the contrastive sampling strategy, the number of pooling iterations ($T$), and the training subset size. Our default configuration is highlighted in bold.

| Component | Setting | mIoU | mAcc |
|---|---|---|---|
| Geometric Purification | w/o Purification (Aggregated 2D features) | 50.2 | 68.1 |
| | **+ GeoPurify (Ours)** | **55.1** | **72.5** |
| 2D Semantic Backbone | LSeg | 48.6 | 61.6 |
| | LSeg + GeoPurify | 51.2 | 63.0 |
| Contrastive Sampling | Macro-only | 53.5 | 70.8 |
| | **Hybrid (Ours)** | **55.1** | **72.5** |
| Pooling Iterations ($T$) | $T = 1$ | 52.3 | 70.2 |
| | $T = 6$ | 53.9 | 71.4 |
| | $T = 18$ **(Ours)** | **55.1** | **72.5** |
| | $T = 36$ | 55.1 | 72.4 |
| Training Subset Size | 10 scenes | 54.7 | 72.4 |
| | **20 scenes (Ours)** | **55.1** | **72.5** |
| | 30 scenes | 55.1 | 72.5 |
| | 50 scenes | 55.0 | 72.5 |

**Geometric Purification** We first validate the efficacy of our core module. A baseline that directly aggregates 2D features from the X-Decoder backbone without any geometric refinement achieves 50.2 mIoU. The introduction of our GeoPurify module yields a significant performance increase to **55.1 mIoU** (+4.9 mIoU). This substantial gain underscores our central hypothesis: even with a powerful semantic backbone, projected features suffer from geometric fragmentation, and an explicit purification step is essential to resolve these inconsistencies.

**2D Semantic Backbone** We next analyze the choice of the 2D backbone. While applying GeoPurify to a standard LSeg backbone achieves a respectable 51.2 mIoU, replacing it with the generalist X-Decoder boosts performance to **55.1 mIoU**. This +3.9 mIoU improvement demonstrates that the richer, more associative feature space of the X-Decoder provides a superior semantic substrate for the subsequent geometric refinement process.

**Hybrid Contrastive Sampling** To dissect the purification module, we evaluate our sampling strategy. A variant trained using only macro-negatives results in a 1.6 mIoU performance drop to 53.5 mIoU. This confirms that micro-negatives are the primary mechanism for resolving local geometric ambiguities at object boundaries (e.g., differentiating a chair leg from the floor). Without them, the model learns the global scene layout but fails to disentangle co-located surfaces.

**Pooling Iterations ($T$)** Finally, we analyze the number of pooling iterations, $T$, which directly controls the receptive field of the feature refinement on the geometric graph. Each iteration $F^{(t+1)} = AF^{(t)}$ propagates information by one hop, meaning a point's feature after $T$ steps is influenced by its neighbors up to $T$ hops away. As shown in Table 4, performance steadily improves from $T = 1$ (52.3 mIoU) to $T = 18$ (**55.1 mIoU**). This suggests that a sufficiently large receptive field is necessary to propagate semantic information across entire coherent surfaces, thereby averaging out initial projection noise. However, increasing the iterations further to $T = 36$ leads to a slight performance degradation. This indicates the onset of over-smoothing, a well-known phenomenon where an excessively large receptive field begins to merge features from distinct but adjacent objects, blurring crucial semantic boundaries. Thus, $T = 18$ represents an optimal trade-off, maximizing information propagation within geometrically consistent regions while preserving the semantic distinctiveness between them.

**Training Subset Size** We evaluate the method's sensitivity to data quantity by training on subsets of 10, 20, 30, and 50 scenes. As shown in Table 4, performance improves markedly from 10 to 20

scenes but plateaus thereafter. This trend validates our design principle: unlike methods learning complex semantic mappings, our student network distills class-agnostic geometric affinities (e.g., surface continuity) which are repetitive and universal. Consequently, a small subset (20 scenes) provides sufficient structural diversity for convergence. The observed plateau indicates that our framework successfully recovers the latent structure available to refine the current VLM features, suggesting performance is primarily bounded by the VLM's semantic quality rather than the quantity of geometric training data.

## 5 CONCLUSION

We introduce GeoPurify, a novel framework that resolves the fundamental conflict between semantic richness and geometric coherence in open-vocabulary 3D scene understanding. We posit that the prevailing "segmentation and matching" paradigm is a primary bottleneck and propose a conceptual shift towards "segmentation as understanding." GeoPurify materializes this vision through a data-efficient teacher-student architecture that distills latent geometric structure from noisy, view-aggregated VLM features. By leveraging Geometric Contrastive Distillation on unlabeled 3D scans and enforcing spatial consistency with Geometry-Guided Pooling at inference, our method effectively reconciles the two modalities. Extensive experiments show that GeoPurify achieves state-of-the-art or competitive performance on major 3D benchmarks without any 3D annotations, highlighting its potential as a scalable, foundational approach for annotation-free 3D perception.

### ACKNOWLEDGMENTS

This work is supported by the Central Guidance on Local Science and Technology Development Fund of Shanghai City (YDZX20253100002004), National Key Research and Development Program of China (2025YFF0522500), the Fundamental and Interdisciplinary Disciplines Breakthrough Plan of the Ministry of Education of China (JYB2025XDXM116), and the Fundamental Research Funds for the Central Universities.

### ETHICS STATEMENT

Our research utilizes publicly available academic benchmarks (ScanNetV2, Matterport3D, Scan-Net200) and adheres to their standard usage protocols. We acknowledge that these datasets may contain geographic and cultural biases, which could impact model fairness and generalization across diverse real-world environments. While the technology is intended for beneficial applications like robotics and augmented reality, we recognize the dual-use potential inherent to all advanced perception systems.

### REPRODUCIBILITY STATEMENT

To ensure our work is reproducible, we will publicly release all code and pre-trained models. The core methodology is detailed in Section 3, with evaluation protocols specified in Section 4.1. Appendix A offers a complete implementation guide, detailing network architectures, the exact data subset selection logic, and a comprehensive hyperparameter table.

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

## A    Experimental Setup and Implementation Details

This section provides the implementation details for our experiments to ensure reproducibility. All 3D sparse operations are implemented using the Minkowski Engine (Choy et al., 2019). Our code and pre-trained models will be made publicly available upon publication.

### A.1    Network Architectures

We detail the architectures for the student, teacher, and 2D semantic backbone networks used in our experiments.

**Student Network ($\phi_S$):**   The student network, which we term the `AffinityPredictor`, is a sparse 3D convolutional network that processes point-wise features. The input is a sparse tensor where each voxel contains a 518-dimensional feature vector, formed by concatenating a 512-dim semantic feature from $\Psi_{2D}$ with a 6-dim geometric attribute vector (RGB and surface normal). The architecture comprises three stages:

1. An **input stem** that uses a $3 \times 3 \times 3$ sparse convolution, followed by batch normalization and ReLU, to project the 518-dim input features to a 512-dim hidden space. 2. A **body** composed of four sequential `MinkowskiResBlock` modules. Each block contains two $3 \times 3 \times 3$ sparse convolutions with batch normalization, a residual connection, and a final ReLU activation. 3. A **projection head** that employs a $1 \times 1 \times 1$ sparse convolution to map the 512-dim features to the final 128-dim embedding.

The complete architecture is detailed in Table 5.

Table 5: **Student Network Architecture ($\phi_S$).** The network operates on sparse 3D tensors. The feature dimension is maintained at 512 throughout the body.

| Stage | Operator | Kernel Size | Stride | Channels (In $\to$ Out) |
|---|---|---|---|---|
| Input | ME.MinkowskiConvolution, ME.BatchNorm, ME.ReLU | $3^3$ | 1 | $518 \to 512$ |
| Body | $4 \times$ MinkowskiResBlock | - | - | $512 \to 512$ |
| | ME.MinkowskiConvolution, ME.BatchNorm, ME.ReLU | $3^3$ | 1 | $512 \to 512$ |
| | ME.MinkowskiConvolution, ME.BatchNorm | $3^3$ | 1 | $512 \to 512$ |
| | + (Identity Shortcut) | - | - | - |
| | ME.ReLU | - | - | - |
| Head | ME.MinkowskiConvolution | $1^3$ | 1 | $512 \to 128$ |

**Teacher Network ($\phi_T$):**   The teacher network is a frozen, pre-trained Sonata model (Wu et al., 2025), a 3D self-supervised model built upon the Point Transformer V3 (PTv3) architecture. We employ this model for its high-quality, 3D-native representations, which serve as the source of stable geometric priors for distillation. The efficacy of these features is demonstrated on the Scan-Net benchmark, where they achieve 72.5% mIoU with linear probing, substantially outperforming 2D-lifted features from models like DINOv2 (63.1% mIoU). The original publication further substantiates the robustness of Sonata's features through qualitative analyses that show clear semantic clustering and strong spatial reasoning. We use the officially released checkpoint and refer readers to the original work for complete architectural details.

**2D Semantic Backbone ($\Psi_{2D}$):**   The 2D semantic backbone provides initial semantic priors using a frozen, pre-trained X-Decoder with a DaViT-L backbone (Zou et al., 2023), for which we use the officially released weights. X-Decoder is a general-purpose model trained on diverse tasks (e.g., segmentation, image-text retrieval) to learn a unified visual-linguistic embedding space. We hypothesize that features learned via this multi-modal, multi-task objective provide richer semantic information for 3D scene understanding than those from networks trained solely on a single task like segmentation.

### A.2 DATASET DETAILS AND SUBSET SELECTION

For all experiments, we adhere to the official training, validation, and testing splits for the Scan-NetV2 and Matterport3D datasets to ensure fair comparison with prior work.

---

**Algorithm 1** Principled Scene Subset Selection
___

**Input:** Full set of regions $\mathcal{R}$, desired subset size $K_{subset}$, number of clusters $K_{clusters}$, richness weight $\gamma$.
**Output:** Selected optimal subset of regions $\mathcal{R}_{selected}$.

**Phase 1: Metric Calculation**
1: Let $StatsList \leftarrow []$
2: **for** each region $r \in \mathcal{R}$ **do**
3:     Extract semantic labels l from $r$.
4:     $N_c(r) \leftarrow$ Count of unique labels in l.                              ▷ Semantic Richness
5:     $p(c) \leftarrow$ Proportion of each label $c$ in l.
6:     $H_c(r) \leftarrow -\sum_c p(c) \log_2 p(c)$.                              ▷ Semantic Complexity (Entropy)
7:     $Hist(r) \leftarrow$ Full histogram of labels in l.
8:     Append $\{r, N_c(r), H_c(r), Hist(r)\}$ to $StatsList$.
9: **end for**

**Phase 2: Filtering**
10: $\tilde{N}_c \leftarrow \text{Median}(\{s.N_c \text{ for } s \in StatsList\})$.
11: $\tilde{H}_c \leftarrow \text{Median}(\{s.H_c \text{ for } s \in StatsList\})$.
12: Let $FilteredList \leftarrow [s \text{ for } s \in StatsList \text{ if } s.N_c \geq \tilde{N}_c \text{ and } s.H_c \geq \tilde{H}_c]$.

**Phase 3: Clustering**
13: $F \leftarrow [s.Hist \text{ for } s \in FilteredList]$.                              ▷ Create feature matrix from histograms
14: $ClusterLabels \leftarrow \text{KMeans}(F, K_{clusters})$.
15: **for** $i \leftarrow 1$ to length($FilteredList$) **do**
16:     $FilteredList[i].cluster \leftarrow ClusterLabels[i]$.
17: **end for**

**Phase 4: Scoring**
18: $N_{c,norm} \leftarrow \text{MinMaxScaler}(\{s.N_c \text{ for } s \in FilteredList\})$.
19: $H_{c,norm} \leftarrow \text{MinMaxScaler}(\{s.H_c \text{ for } s \in FilteredList\})$.
20: **for** $i \leftarrow 1$ to length($FilteredList$) **do**
21:     $FilteredList[i].score \leftarrow H_{c,norm}[i] + \gamma \cdot N_{c,norm}[i]$.
22: **end for**

**Phase 5: Stratified Selection**
23: Let $\mathcal{R}_{selected} \leftarrow []$
24: $n_{base} \leftarrow \lfloor K_{subset}/K_{clusters} \rfloor$
25: $n_{rem} \leftarrow K_{subset} \pmod{K_{clusters}}$
26: **for** $j \leftarrow 0$ to $K_{clusters} - 1$ **do**
27:     $Cluster_j \leftarrow \{s \text{ for } s \in FilteredList \text{ if } s.cluster = j\}$.
28:     Sort $Cluster_j$ in descending order by $s.score$.
29:     $n_{select} \leftarrow n_{base} + (1 \text{ if } j < n_{rem} \text{ else } 0)$.
30:     Add top $n_{select}$ regions from sorted $Cluster_j$ to $\mathcal{R}_{selected}$.
31: **end for**

32: **return** $\mathcal{R}_{selected}$.

---

To substantiate our claims of data efficiency, we select a compact training subset using the principled, automated pipeline detailed in Algorithm 1. This method is configured with key hyperparameters. To align with the categorical diversity of each dataset, the number of clusters $K$ is set based on the variety of real-world spaces documented in the original papers: we use $K = 20$ **for ScanNetV2** and $K = 12$ **for Matterport3D**. The richness-complexity score weight is held constant at $\gamma = 0.5$. This process yields a final subset of 20 scenes and 20 regions, respectively. The complete lists of the selected identifiers are provided in Table 6.

Table 6: Final scene and region IDs constituting our compact training subsets for ScanNetV2 and Matterport3D.

| ScanNetV2 Selected Scenes | Matterport3D Selected Regions |
|---|---|
| scene0126_00 | VzqfbhrpDEA_region7 |
| scene0640_02 | cV4RVeZvu5T_region11 |
| scene0247_01 | 759xd9YjKW5_region12 |
| scene0604_00 | ULsKaCPVFJR_region17 |
| scene0315_00 | uNb9QFRL6hY_region33 |
| scene0604_02 | vyrNrziPKCB_region57 |
| scene0102_01 | D7N2EKCX4Sj_region11 |
| scene0137_01 | 759xd9YjKW5_region16 |
| scene0588_03 | mJXqzFtmKg4_region2 |
| scene0547_01 | EDJbREhghzL_region3 |
| scene0451_02 | VLzqgDo317F_region8 |
| scene0341_00 | VLzqgDo317F_region9 |
| scene0106_00 | VzqfbhrpDEA_region2 |
| scene0592_00 | B6ByNegPMKs_region8 |
| scene0692_00 | D7N2EKCX4Sj_region41 |
| scene0393_00 | D7N2EKCX4Sj_region38 |
| scene0340_00 | 2n8kARJN3HM_region13 |
| scene0220_00 | b8cTxDM8gDG_region14 |
| scene0394_00 | VFuaQ6m2Qom_region12 |
| scene0692_01 | 82sE5b5pLXE_region5 |

### A.3 COMPREHENSIVE HYPERPARAMETER TABLE

To ensure full reproducibility of our experimental results, we provide a comprehensive list of all hyperparameters used for model training, distillation, and inference. Our final configuration is detailed in Table 7.

## B ADDITIONAL QUANTITATIVE RESULTS

This section provides a detailed breakdown of our method's performance to supplement the summary results in the main paper and offer concrete evidence for our core technical contributions.

### B.1 PER-CLASS SEGMENTATION RESULTS

The per-class IoU scores in Tables 8 and 9 offer a granular validation of our central hypothesis: that explicitly purifying the geometric structure of projected 2D features is critical for robust 3D segmentation. The baseline, which relies directly on multi-view aggregated features from the X-Decoder VLM, suffers from the geometric inconsistencies inherent to view projection, leading to fragmented and noisy object representations. Our method, GeoPurify, systematically addresses this deficiency.

On ScanNetV2 (Table 8), GeoPurify improves mIoU by **4.9%**. The most substantial gains are observed in categories where object identity is strongly tied to geometric structure rather than just texture or color. For instance, classes with intricate parts and thin structures, such as chair (+8.5%), sofa (+7.1%), and desk (+5.6%), see dramatic improvements. This is a direct result of our Geometry-Guided Pooling mechanism. The baseline often fractures the features of a chair's legs and backrest, mixing them with features from the background. In contrast, our contrastively trained student network learns to establish high geometric affinity among all points belonging to the chair's structure. The subsequent pooling step then uses this affinity graph to enforce semantic consistency, effectively "denoising" the initial VLM features and producing a coherent object representation.

A similar pattern emerges on the more complex Matterport3D benchmark (Table 9), where our method attains a **2.7%** higher mIoU. The significant improvements for classes like toilet (+6.8%)

Table 7: Comprehensive list of hyperparameters. All settings are kept consistent across datasets unless otherwise specified.

| Hyperparameter | Value |
|---|---|
| **Training Parameters** | |
| Optimizer | AdamW |
| Base Learning Rate ($lr_{base}$) | 1e-4 |
| Learning Rate Scaling | Input layer: $0.1 \times lr_{base}$ |
| | Middle layers: $1.0 \times lr_{base}$ |
| | Output layer: $5.0 \times lr_{base}$ |
| LR Scheduler | Cosine Annealing with Linear Warmup |
| Warmup Epochs | 2 |
| Warmup Start Factor | 1e-6 |
| Cosine Annealing Min LR ($\eta_{min}$) | 1e-7 ($10^{-3} \times lr_{base}$) |
| Weight Decay | 1e-5 |
| Total Epochs | 50 |
| **Contrastive Distillation** | |
| InfoNCE Temperature ($\tau$) | 0.07 |
| Anchor Points (per scene) | 4096 |
| Macro-negatives (per anchor) | 48 |
| Micro-negatives (per anchor) | 16 |
| **Inference & Pooling** | |
| Pooling Iterations ($T$) | 18 |
| Affinity Sharpening Factor ($\alpha$) | 0.05 (1/20.0) |
| Nearest Neighbors for Affinity ($K$) | 96 |
| Voxel Size | 0.02 m |

and curtain (+5.9%) further validate our approach. These objects often have uniform textures where 2D appearance can be ambiguous; their defining characteristic is their 3D shape. GeoPurify excels here because our geometry-centric student network provides the precise structural cues needed to group points correctly, overcoming the semantic ambiguity of the initial 2D features.

Conversely, the results also highlight the current limitations of our local pooling strategy. Classes defined by extremely thin geometry and a strong co-planar relationship with a larger surface, such as a picture and a shower curtain, show minimal gains or even slight degradation. In these scenarios, the local neighborhood queried by our pooling mechanism is dominated by the larger surface (e.g., a wall). Consequently, the powerful semantic features of the wall can "leak" into the picture's features during the affinity-based aggregation, overwhelming its distinct identity. This indicates that while our method robustly handles complex object-level geometry, resolving features for extremely fine-grained, surface-level details remains a challenge for future work.

Table 8: Per-class IoU (%) on the ScanNetV2 validation set. **mIoU**: Baseline 50.2, GeoPurify (Ours) **55.1** (+4.9). **mAcc**: Baseline 68.1, GeoPurify (Ours) **72.5** (+4.4).

| Method | wall | floor | cabinet | bed | chair | sofa | table | door | window | bookshelf | picture | counter | desk | curtain | fridge | shower c. | toilet | sink | bathtub |
|---|---|---|---|---|---|---|---|---|---|---|---|---|---|---|---|---|---|---|---|
| Baseline | 72.0 | 76.4 | 46.4 | 66.7 | 57.1 | 64.5 | 44.0 | 54.6 | 57.3 | 65.6 | 1.7 | 28.5 | 30.8 | 50.6 | 42.4 | 39.8 | 57.9 | 37.5 | 59.2 |
| GeoPurify (Ours) | **75.0** | **81.4** | **50.9** | **70.8** | **65.7** | **71.6** | **51.8** | **59.0** | **62.4** | **68.9** | 0.0 | **30.9** | **36.4** | **54.8** | **49.6** | **44.8** | **66.8** | **42.3** | **64.6** |
| Δ | +3.0 | +5.0 | +4.5 | +4.1 | +8.5 | +7.1 | +7.8 | +4.4 | +5.1 | +3.3 | -1.7 | +2.4 | +5.6 | +4.2 | +7.2 | +5.0 | +8.9 | +4.8 | +5.4 |

Table 9: Per-class IoU (%) on the Matterport3D test set. **mIoU**: Baseline 37.5, GeoPurify (Ours) **40.2** (+2.7). **mAcc**: Baseline 59.8, GeoPurify (Ours) **62.4** (+2.6).

| Method | wall | floor | cabinet | bed | chair | sofa | table | door | window | bookshelf | picture | counter | desk | curtain | fridge | shower c. | toilet | sink | bathtub | ceiling |
|---|---|---|---|---|---|---|---|---|---|---|---|---|---|---|---|---|---|---|---|---|
| Baseline | 63.8 | 71.7 | 37.7 | 67.1 | 55.2 | 51.3 | 26.0 | 43.8 | 38.9 | **11.5** | **0.8** | **16.6** | 15.9 | 43.4 | 25.0 | 6.8 | 41.5 | 26.1 | 33.3 | 74.0 |
| GeoPurify (Ours) | **65.5** | **73.3** | **39.1** | **71.3** | **60.6** | **54.7** | **28.2** | **46.7** | **42.1** | 11.1 | 0.1 | 16.0 | **19.8** | **49.3** | **27.7** | **7.6** | **48.3** | **27.3** | **39.0** | **76.4** |
| Δ | +1.7 | +1.6 | +1.4 | +4.2 | +5.4 | +3.4 | +2.2 | +2.9 | +3.2 | -0.4 | -0.7 | -0.6 | +3.9 | +5.9 | +2.7 | +0.8 | +6.8 | +1.2 | +5.7 | +2.4 |

### B.2 CROSS-DATASET GENERALIZATION

As demonstrated in the main text, our decoupled geometric prior provides robust generalization across different 3D domains. This section presents a more granular analysis to further probe the limits of this robustness, subjecting the model to a simultaneous shift in both geometric distribution and semantic vocabulary size. To this end, we evaluate a model trained on ScanNetV2 on increasingly larger subsets of the Matterport3D dataset, defined by the $K$ most common categories for $K \in \{40, 80, 160\}$.

The results, shown in Table 10, reveal that the performance advantage of GeoPurify over competing methods magnifies as the vocabulary expands. This trend starkly illustrates the failure mode of models with entangled geo-semantic representations. When confronted with a large set of unfamiliar classes, their geometric reasoning falters, as it is implicitly conditioned on the semantic distribution of the source domain. This leads to a rapid degradation in performance as vocabulary size increases.

In contrast, the superior scaling of our method validates the benefit of a decoupled architecture. Because GeoPurify's student network learns a purely structural, class-agnostic prior, its ability to enforce geometric coherence is not compromised by the novelty or diversity of the semantic features. It continues to function as a powerful regularizer, denoising the initial 2D features and preserving structural integrity. This confirms that our method's robustness is a fundamental property that scales effectively to more challenging and diverse open-world scenarios.

Table 10: Comparison on **cross-dataset** generalization against state-of-the-art methods. All models are trained on ScanNetV2 and tested zero-shot on Matterport3D. We evaluate on the original 21-class benchmark and on subsets containing the $K$ most common categories from the NYUv2 label set, where $K \in \{40, 80, 160\}$.

| Method | Matterport21 | | Matterport40 | | Matterport80 | | Matterport160 | |
|---|---|---|---|---|---|---|---|---|
| | mIoU | mAcc | mIoU | mAcc | mIoU | mAcc | mIoU | mAcc |
| OpenScene (Peng et al., 2023) | 36.0 | 48.0 | 21.1 | 27.5 | 10.8 | 13.9 | 6.0 | 8.1 |
| CUA-O3D (Li et al., 2025) | 37.4 | 49.2 | 23.3 | 30.2 | 12.2 | 16.3 | 6.1 | 8.4 |
| GGSD (Wang et al., 2024b) | 40.1 | 54.4 | 22.8 | 31.6 | 11.9 | 16.2 | 6.3 | 9.6 |
| **GeoPurify (Ours)** | **40.5** | **62.7** | **33.3** | **48.3** | **19.1** | **32.8** | **12.3** | **21.6** |

## C SUBSET SELECTION STRATEGY

To ensure that the efficacy of GeoPurify is not an artifact of the specific data sampling method, we evaluate its robustness against two fully annotation-free alternatives: random uniform selection and unsupervised clustering based on spatial and colorimetric features. As reported in Table 11, the model trained on randomly selected scenes achieves an mIoU of 54.6%, representing a negligible deviation from the heuristic baseline. Similarly, the unsupervised clustering strategy yields 54.9% mIoU. These findings confirm that the observed performance gains are intrinsic to the proposed geometric purification mechanism and do not rely on specific high-quality samples or global statistical guidance.

Table 11: **Robustness analysis of subset selection strategies on ScanNetV2 ($\sim$1.5% data).** The comparison demonstrates that GeoPurify maintains consistent performance across varying selection methodologies.

| Selection Strategy | mIoU | mAcc |
|---|---|---|
| Heuristic (Ours) | 55.1 | 72.5 |
| Random | 54.6 | 72.2 |
| Coordinate and Color Clustering | 54.9 | 72.3 |

# D    EXTENDED QUALITATIVE ANALYSIS

This section provides a detailed qualitative analysis of our method on the ScanNetV2 and Matterport3D datasets to offer a comprehensive understanding of its capabilities and limitations.

## D.1    VISUALIZATION OF INTERMEDIATE MECHANISMS

Figure 3 provides a conceptual illustration of our Geometry-Guided Pooling module, which is mathematically detailed in the main text. This mechanism is designed to rectify the geometric inconsistencies present in the initial semantic features that are aggregated from multi-view 2D images.

The process begins with the trained student network, $\phi_S$, which generates a distinct geometric embedding for each point in the scene. From these embeddings, we construct a sparse affinity matrix that encodes structural relationships within local neighborhoods. The strength of the affinity between two neighboring points is derived from the sharpened cosine similarity of their embeddings, concentrating the connection weights onto the most structurally similar points—for instance, those lying on the same continuous surface (visualized as green lines). The purification itself is an iterative refinement, where the semantic features are updated at each step by being multiplied with this affinity matrix. This operation functions as a geometry-aware filter, effectively propagating and averaging semantic information across structurally coherent regions. As depicted, this process denoises the initial fragmented features, yielding a final representation that is both semantically rich and geometrically sound.

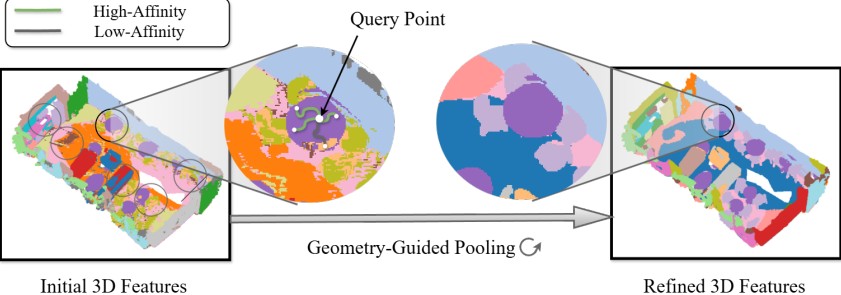

Figure 3: **The Geometry-Guided Pooling Mechanism.** *Left:* Initial point cloud features aggregated from 2D views exhibit significant fragmentation and geometric inconsistency. *Center:* Our pooling mechanism operates locally. For a given query point, the learned affinity matrix identifies nearby points on the same surface as having high affinity (green lines) while assigning low affinity to points on other surfaces (gray lines). *Right:* By iteratively averaging features among high-affinity neighbors, the process denoises the initial features, producing a final output that is both semantically rich and geometrically coherent.

## D.2    QUALITATIVE RESULTS

We first provide a visual analysis of feature coherence before and after our proposed purification. As illustrated in Figure 4, initial features aggregated from 2D views exhibit significant fragmentation. Visualized via k-means clustering, these features yield scattered, incoherent segments that do not align with the underlying object geometry. In contrast, after applying GeoPurify's Geometry-Guided Pooling, the refined features produce contiguous clusters that precisely match distinct surfaces, such as walls, floors, and furniture. This result demonstrates our method's effectiveness in instilling structural awareness into the 3D feature space.

Next, we evaluate the impact of this enhanced feature quality on open-vocabulary semantic segmentation. Figure 5 presents a side-by-side comparison in challenging scenes from the ScanNetV2 validation and Matterport3D test sets. The baseline, which uses unpurified features, generates noisy predictions with inaccurate boundaries and fragmented object detections. GeoPurify consistently mitigates these artifacts, producing segmentation maps that are substantially more coherent and pre-

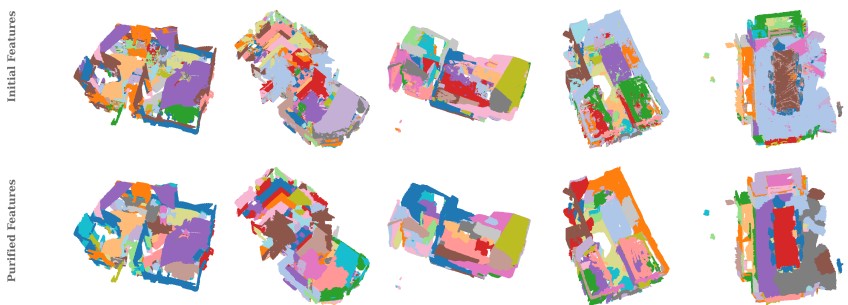

Figure 4: **Visualization of Feature Coherence.** *Point clouds are colored by applying k-means clustering to their features.* **Top Row (Initial Features):** *Features aggregated directly from 2D views lead to fragmented and geometrically inconsistent clusters.* **Bottom Row (Purified Features):** *Our Geometry-Guided Pooling produces smooth, coherent features that align with the underlying 3D structure.*

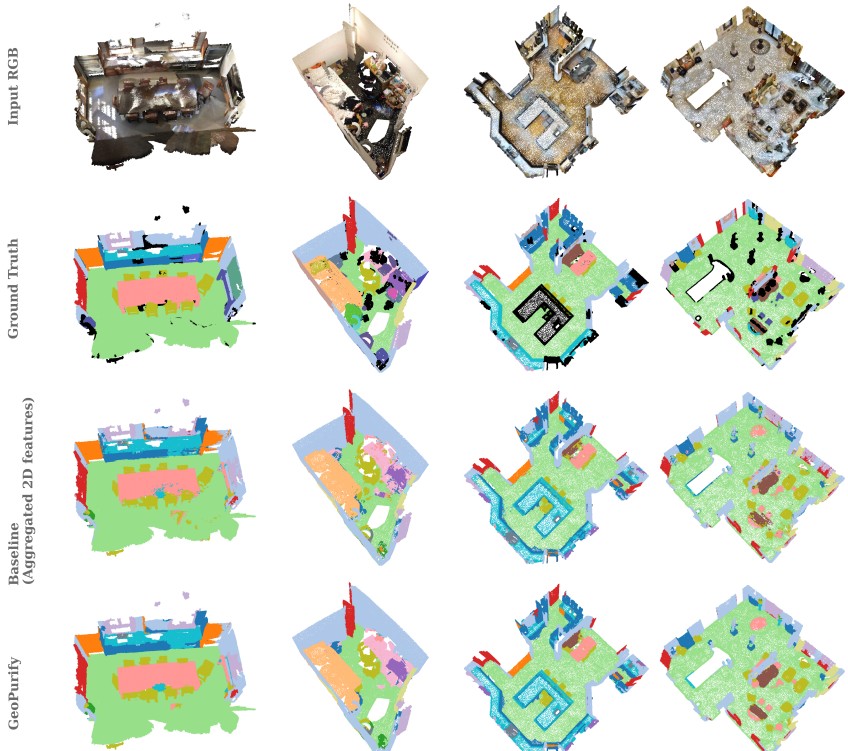

Figure 5: **Qualitative Comparison on ScanNetV2 and Matterport3D.** *Visual results on challenging indoor scenes. From top to bottom: Input RGB point cloud, Ground Truth segmentation, Baseline predictions (from aggregated 2D features), and our **GeoPurify** predictions. Our method generates significantly cleaner segmentation maps with sharper boundaries and more complete object segments compared to the baseline.*

cise. Our method successfully delineates sharp boundaries and captures object instances with greater completeness, yielding results that more closely approximate the ground truth.

### D.3 FAILURE CASE ANALYSIS

To provide a transparent assessment, we analyze the primary failure modes of our model, as shown in Figure 6. We identify two principal categories of errors:

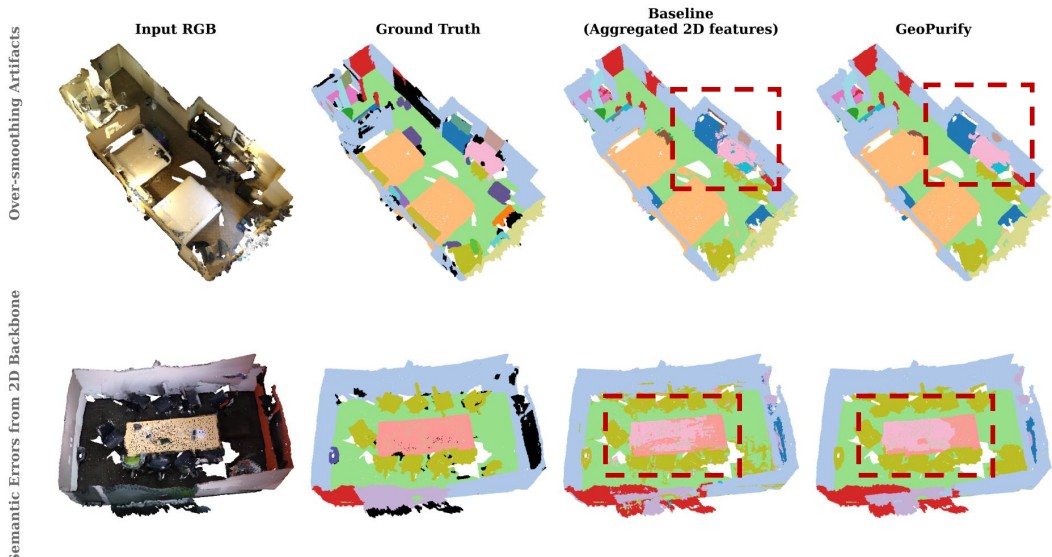

Figure 6: **Illustration of typical failure modes.** From left to right: challenges with the presence of over-smoothing artifacts at object boundaries, and inherited semantic errors from the initial 2D feature backbone.

- **Over-smoothing Artifacts:** In some instances, the iterative pooling process can subtly blur the boundaries between distinct but closely adjacent objects, resulting in minor local inaccuracies.
- **Semantic Errors from 2D Backbone:** The performance of GeoPurify is dependent on the quality of the initial 2D features. When the 2D backbone generates a fundamental semantic error, our geometric purification module is often unable to correct the misclassification.

These limitations highlight key challenges and suggest promising directions for future research, including the development of more robust feature fusion techniques and error correction mechanisms.

## E  THE USE OF LARGE LANGUAGE MODELS

A large language model (LLM) was utilized as a writing assistant in the preparation of this manuscript. The scope of its use was strictly limited to article polishing, including improving grammar, enhancing clarity, and ensuring conciseness. All core scientific contributions, such as research ideation, methodology development, and experimental analysis, were conceived and conducted by the human authors. The authors have reviewed all LLM-generated text and take full responsibility for the scientific accuracy and integrity of the final content.

## APPENDIX REFERENCES

Christopher Choy, JunYoung Gwak, and Silvio Savarese. 4d spatio-temporal convnets: Minkowski convolutional neural networks. In *Proceedings of the IEEE/CVF conference on computer vision and pattern recognition*, pp. 3075–3084, 2019.

Jinlong Li, Cristiano Saltori, Fabio Poiesi, and Nicu Sebe. Cross-modal and uncertainty-aware agglomeration for open-vocabulary 3d scene understanding. In *Proceedings of the Computer Vision and Pattern Recognition Conference*, pp. 19390–19400, 2025.

Songyou Peng, Kyle Genova, Chiyu Jiang, Andrea Tagliasacchi, Marc Pollefeys, Thomas Funkhouser, et al. Openscene: 3d scene understanding with open vocabularies. In *Proceedings of the IEEE/CVF conference on computer vision and pattern recognition*, pp. 815–824, 2023.

Pengfei Wang, Yuxi Wang, Shuai Li, Zhaoxiang Zhang, Zhen Lei, and Lei Zhang. Open vocabulary 3d scene understanding via geometry guided self-distillation. In *European Conference on Computer Vision*, pp. 442–460. Springer, 2024.

Xiaoyang Wu, Daniel DeTone, Duncan Frost, Tianwei Shen, Chris Xie, Nan Yang, Jakob Engel, Richard Newcombe, Hengshuang Zhao, and Julian Straub. Sonata: Self-supervised learning of reliable point representations. In *Proceedings of the Computer Vision and Pattern Recognition Conference*, pp. 22193–22204, 2025.

Xueyan Zou, Zi-Yi Dou, Jianwei Yang, Zhe Gan, Linjie Li, Chunyuan Li, Xiyang Dai, Harkirat Behl, Jianfeng Wang, Lu Yuan, et al. Generalized decoding for pixel, image, and language. In *Proceedings of the IEEE/CVF conference on computer vision and pattern recognition*, pp. 15116–15127, 2023.

