# OpenReview forum: "GeoPurify: A Data-Efficient Geometric Distillation Framework for Open-Vocabulary 3D Segmentation"
_ICLR.cc/2026/Conference — ICLR 2026 Poster_

### Official Review · Reviewer_1k2F · 2025-10-27

**Soundness:** 3
**Presentation:** 2
**Contribution:** 2
**Rating:** 6
**Confidence:** 3

**Summary:**

This paper proposes an open-vocabulary 3D segmentation framework, GeoPurify, which purifies noisy semantic features using 3D geometric guidance. The approach introduces a lightweight Student Affinity Network that learns pure geometric relationships via a Geometric Contrastive Distillation scheme, without requiring any 3D semantic labels. The method demonstrates notable data efficiency, as using only ~1.5% of the training scenes achieves comparable performance to state-of-the-art methods trained on the full dataset.

**Strengths:**

1. The method achieves SOTA performance using ~1.5% of unlabeled 3D scan data, significantly reducing reliance on costly 3D annotations, offering a practical and promising solution for 3D understanding in data-scarce scenarios.
2. This paper learns a pure geometric prior to regularize and enhance semantic features, which addresses the key issue of geometric fragmentation resulting from the 2D-to-3D projection.

**Weaknesses:**

1. The choice of 20 scenes lacks empirical justification. An ablation study examining performance trends across different subset sizes (e.g., 10, 30, or 50 scenes) is missing, making it difficult to assess the optimality of the selected size or the method's sensitivity to training data quantity.
2. Would there be more analysis on the effectiveness of data efficiency? Table 1 presents the severe degradation of CUA-O3D under a ~1.5% subset. Why does the proposed method not show such a degradation?
3. Lack of discussions with related methods. Recent state-of-the-art methods, such as Open3DIS [1] and UniSeg3D [2], have reported results on ScanNet200, which are not discussed in this paper. Are the experimental settings different?

[1] Open3DIS: Open-Vocabulary 3D Instance Segmentation with 2D Mask Guidance. CVPR 24.

[2] A Unified Framework for 3D Scene Understanding. NeurIPS 24.

**Questions:**

Please refer to details in the above section.

---

> ### Author Response · Authors · 2025-11-21
>
> We appreciate the reviewer 1k2F recognition of our method's data efficiency and its ability to address geometric fragmentation. We address the specific concerns regarding the training subset size, the mechanism of data efficiency, and related works below.
>
> > W1:The choice of 20 scenes lacks empirical justification. An ablation study examining performance trends across different subset sizes (e.g., 10, 30, or 50 scenes) is missing, making it difficult to assess the optimality of the selected size or the method's sensitivity to training data quantity.
>
> We have addressed the concern regarding the training subset size by conducting the suggested ablation study with 10, 20, 30, and 50 scenes. As shown in the table below, performance improves noticeably from 10 to 20 scenes but plateaus thereafter, with negligible differences between 20, 30, and 50 scenes.
>
> | Training Subset Size | mIoU (%) | mAcc (%) |
> | :------------------: | :------: | :------: |
> |      10 scenes       |   54.7   |   72.4   |
> |    **20 scenes**     | **55.1** | **72.5** |
> |      30 scenes       |   55.1   |   72.5   |
> |      50 scenes       |   55.0   |   72.5   |
>
> This trend validates our design principle: unlike methods that must learn complex semantic mappings from scratch, **our Student Affinity Network is tasked only with distilling "class-agnostic geometric affinities" (e.g., surface continuity, adjacency).** Because these local geometric primitives are repetitive and universal across indoor environments, a subset of 20 scenes provides sufficient structural diversity for the network to converge. Crucially, the observed performance plateau suggests that our framework has successfully recovered the latent geometric structure available to refine the current VLM features. This indicates a desirable property: our method’s performance is correlated with the semantic quality of the VLM backbone. As VLM architectures evolve, our data-efficient framework can be readily applied to these improved models, allowing for sustainable performance gains with minimal training cost. We will include these results in the Section 4.3 to empirically justify our selection.
>
> > W2:Would there be more analysis on the effectiveness of data efficiency? Table 1 presents the severe degradation of CUA-O3D under a ~1.5% subset. Why does the proposed method not show such a degradation?
>
> The stark contrast in degradation between CUA-O3D and GeoPurify under the low-data regime stems from our core hypothesis: **transferring 2D features into 3D does not destroy geometric information but renders it latent.** Standard methods like CUA-O3D typically attempt to learn entangled geo-semantic representations from scratch. In data-scarce scenarios (~1.5%), these methods fail to generalize because they lack sufficient examples to establish robust decision boundaries for high-dimensional semantic features. In contrast, GeoPurify decouples the problem. **We rely on the frozen VLM for semantics and train the student network solely to recover the _latent_ geometric structure (i.e., deciding which points belong together) via contrastive distillation.** Learning these geometric grouping rules requires significantly less data than learning semantic classifications, allowing our method to function as an efficient "geometric denoiser" that maintains high performance even when training data is severely limited. We will incorporate additional analysis and discussion in Section 4.2 of the submission.
>
> > W3: Lack of discussions with related methods. Recent state-of-the-art methods, such as Open3DIS [1] and UniSeg3D [2], have reported results on ScanNet200, which are not discussed in this paper. Are the experimental settings different?
>
> We acknowledge the relevance of Open3DIS and UniSeg3D as recent state-of-the-art approaches in open-vocabulary 3D understanding. However, a direct quantitative comparison in our current tables is not feasible due to a fundamental difference in tasks and metrics. Our work focuses on 3D Semantic Segmentation, evaluated using mean Intersection-over-Union (mIoU) and mean Accuracy (mAcc). In contrast, the results reported for Open3DIS and UniSeg3D on ScanNet200 typically focus on 3D Instance Segmentation, evaluated using Average Precision (AP). As these metrics measure different objectives-region-level classification versus instance-level detection and separation-direct comparison would be misleading. Nevertheless, we recognize the importance of these works in the broader landscape of 3D scene understanding. We will update our Related Work section to discuss these methods, highlighting their contributions to instance-level tasks.

---

> ### Author Response · Authors · 2025-11-27
>
> Dear Reviewer 1k2F,
>
> Thank you once again for your insightful and constructive review, which has greatly contributed to improving the quality and clarity of our paper Geopurify. We sincerely hope that our rebuttal has satisfactorily addressed your questions and concerns.
>
> If you have any remaining questions or would like further clarification, we would be happy to provide it. We truly appreciate your valuable feedback and the time and effort you dedicated to reviewing our work.
>
> Best regards,
>
> Authors of Submission 1462

---

### Official Review · Reviewer_MQX3 · 2025-10-30

**Soundness:** 3
**Presentation:** 3
**Contribution:** 3
**Rating:** 8
**Confidence:** 4

**Summary:**

This paper presents GeoPurify, a data-efficient geometric distillation framework for open-vocabulary 3D segmentation. The work identifies a fundamental disconnect between semantic richness and geometric coherence in existing methods and proposes a shift from the "segmentation and matching" paradigm to "segmentation as understanding." The core idea is to leverage a student affinity network to distill latent geometric structure from noisy 2D VLM-generated 3D features under the guidance of a self-supervised 3D teacher model. A geometry-guided pooling mechanism is further introduced to refine features during inference. The method is evaluated on multiple benchmarks and shows competitive or state-of-the-art performance using only ~1.5% of the training data, demonstrating strong data efficiency and generalization ability.

**Strengths:**

1. The paper clearly articulates the limitations of existing approaches in reconciling 2D semantics with 3D geometry and proposes a principled shift in perspective toward "segmentation as understanding." The idea of recovering latent geometric information from noisy 2D features, rather than learning 3D geometry from scratch, is both novel and well-motivated.

2. The paper provides extensive evaluations across multiple datasets. The comparison with a strong baseline under the same low-data regime is particularly compelling.

**Weaknesses:**

1. Some figures contain low-resolution text and should be improved. The authors should consider replacing these with higher-resolution images or vector graphics to improve readability.

2. Heavy reliance on pre-trained models; the contribution breakdown is unclear. The performance gains of GeoPurify are built upon powerful pre-trained models. While the framework itself is novel, it remains unclear how much of the improvement comes from the proposed distillation and pooling mechanisms versus the strength of the pre-trained backbones.

**Questions:**

This paper presents a well-designed framework that effectively integrates contrastive distillation and geometric pooling, but it fails to clearly demonstrate how its core idea of learning geometric affinity structures provides advantages beyond existing methods.

---

> ### Author Response · Authors · 2025-11-21
>
> We appreciate the reviewer MQX3 assessment and the recognition of our framework’s data efficiency and generalization capabilities. We address the specific concerns raised below.
>
> > W1:Some figures contain low-resolution text and should be improved. The authors should consider replacing these with higher-resolution images or vector graphics to improve readability.
>
> We acknowledge the issue regarding low-resolution text in certain figures. We will update the manuscript to include high-resolution vector graphics, ensuring that all visual details and text in the pipeline overview and qualitative comparisons are fully readable.
>
> > W2:Heavy reliance on pre-trained models; the contribution breakdown is unclear. The performance gains of GeoPurify are built upon powerful pre-trained models. While the framework itself is novel, it remains unclear how much of the improvement comes from the proposed distillation and pooling mechanisms versus the strength of the pre-trained backbones.
>
> Regarding the reliance on pre-trained models, we emphasize that our contribution lies in the methodology of integrating these backbones effectively, rather than merely their selection. As noted in the submission, directly projecting features from even a powerful 2D VLM yields noisy and fragmented predictions due to a lack of 3D spatial awareness; our ablation study confirms this (See Table 4 in the submission), showing that the baseline using X-Decoder features without our framework achieves only 50.2 mIoU on ScanNetV2. **Our proposed "Geometric Purification" module specifically targets these inconsistencies, boosting performance to 55.1 mIoU through our distillation and pooling mechanisms.** To further demonstrate that these gains are not solely dependent on a specific powerful backbone, **we applied GeoPurify to a standard LSeg backbone, which also resulted in a performance increase from 48.6 to 51.2 mIoU.** This indicates that the "Geometric Contrastive Distillation" effectively refines representations regardless of the semantic initialization, validating the framework's independent contribution.
>
> > Q:This paper presents a well-designed framework that effectively integrates contrastive distillation and geometric pooling, but it fails to clearly demonstrate how its core idea of learning geometric affinity structures provides advantages beyond existing methods.
>
> We acknowledge that the initial manuscript did not sufficiently explicate _how_ this core idea provides a competitive edge over existing methods, and we have revised the introduction and contribution sections to clarify this rationale. The primary advantage of our approach stems from the hypothesis that **VLM-projected features, despite being noisy, retain a _latent_ 3D geometric structure.** Unlike existing methods that often attempt to learn 3D geometry and semantics simultaneously from scratch—a process that is highly data-intensive—GeoPurify focuses on _recovering_ this latent structure. By training the Student Affinity Network to learn geometric relationships (affinities) from unlabeled scans, we effectively decouple the semantic richness of the VLM from the structural consistency of the 3D data. This learned affinity structure serves as a robust, class-agnostic geometric prior for our "Geometry-Guided Pooling" module, which denoises fragmented VLM features by propagating semantic information exclusively across physically coherent regions. Furthermore, because these affinities are distilled purely from geometry, they are immune to the long-tail frequency biases inherent in semantic datasets. This structural independence allows GeoPurify to achieve competitive or state-of-the-art performance using only **~1.5%** of the training data, demonstrating superior data efficiency and generalization capabilities compared to methods relying on dense supervision.

---

> ### Author Response · Authors · 2025-11-27
>
> Dear Reviewer MQX3,
>
> Thank you once again for your insightful and constructive review, which has greatly contributed to improving the quality and clarity of our paper Geopurify. We sincerely hope that our rebuttal has satisfactorily addressed your questions and concerns.
>
> If you have any remaining questions or would like further clarification, we would be happy to provide it. We truly appreciate your valuable feedback and the time and effort you dedicated to reviewing our work.
>
> Best regards,
>
> Authors of Submission 1462

---

### Official Review · Reviewer_tyeq · 2025-10-31

**Soundness:** 2
**Presentation:** 3
**Contribution:** 2
**Rating:** 6
**Confidence:** 4

**Summary:**

The paper introduces GeoPurify, a method for open-vocabulary 3D semantic segmentation that enhances the 3D consistency of features extracted from 2D vision-language foundation models. The approach first extracts 2D features using X-Decoder and merges them in 3D via a student affinity network trained to learn geometric relationships by distilling knowledge from the 3D foundation model Sonata (Wu et al.). Training is performed on only a small subset of ScanNetV2 (1.6%) and Matterport3D (1.3%), demonstrating data efficiency. Evaluation on ScanNetV2, Matterport3D, and ScanNet200 shows that GeoPurify outperforms training-based baselines trained on similar subsets, and performs competitively-though slightly worse-than full-data and zero-shot baselines.

**Strengths:**

1. **Effective use of Sonata for geometric distillation.** The paper presents an interesting idea of enhancing the 3D consistency of 2D foundation model features by distilling geometric relationships from a strong 3D foundation model (Sonata). This cross-modal distillation strategy is conceptually sound and well motivated.
2. **Clear and well-structured presentation.** The paper is generally clearly written and easy to follow.

**Weaknesses:**

1. **Limited performance gains.** Although the method is conceptually interesting, it does not clearly outperform strong zero-shot baselines. This raises uncertainty about the practical benefit of the proposed geometric distillation mechanism.
2. **Restricted evaluation scope.** The method is only evaluated in a low-data regime. Assessing performance when trained on larger portions of the dataset would provide a clearer picture of its scalability and potential advantages over existing approaches

**Questions:**

1. Would it be possible to elaborate on why in Table 1 GeoPurify does not improve much over the zero shot baselines?
2. I believe the method can be interesting even if trained on more data. Therefore, it would be interesting to evaluate its performances when trained on the whole datasets and see whether in that way it can improve over the other baselines.
3. Is the point cloud required as input at inference time? From the text it seems to be the case (line 155), but from Figure 2 it seems to be needed only at training time.

---

> ### Author Response · Authors · 2025-11-21
>
> We appreciate the reviewer tyeq assessment of our cross-modal distillation strategy and the recognition of the method's conceptual soundness. We address the specific concerns regarding performance gains, scalability, and inference requirements below.
>
> > W1 & Q1: Limited performance gains. Although the method is conceptually interesting, it does not clearly outperform strong zero-shot baselines. This raises uncertainty about the practical benefit of the proposed geometric distillation mechanism. Would it be possible to elaborate on why in Table 1 GeoPurify does not improve much over the zero shot baselines?
>
> GeoPurify achieves performance that is competitive with or superior to state-of-the-art zero-shot baselines trained on full datasets. Specifically, on ScanNetV2, our method achieves 55.1 mIoU, surpassing both OpenScene-3D (51.6 mIoU) and CUA-O3D (54.1 mIoU). However, **the primary contribution of GeoPurify is its ability to achieve extreme data efficiency with the recovery of latent geometric structures rather than solely aiming for peak performance through massive supervision.** The practical value is best highlighted when comparing methods under similar data constraints; for instance, when the strong baseline CUA-O3D is restricted to the same ~1.5% data subset, its performance drops significantly to 18.1 mIoU, whereas GeoPurify maintains 55.1 mIoU. This validates that our geometric distillation mechanism effectively recovers latent structure where other methods fail without large-scale data. Furthermore, in the challenging cross-dataset transfer from Matterport3D to ScanNetV2, GeoPurify achieves 54.9 mIoU, surpassing the next best method by 16.3 points. This indicates that while our in-domain metrics are competitive, the geometric priors learned are significantly more robust and generalized than those of existing baselines.
>
> > W2 & Q2: Restricted evaluation scope. The method is only evaluated in a low-data regime. Assessing performance when trained on larger portions of the dataset would provide a clearer picture of its scalability and potential advantages over existing approaches. I believe the method can be interesting even if trained on more data. Therefore, it would be interesting to evaluate its performances when trained on the whole datasets and see whether in that way it can improve over the other baselines.
>
> The reviewer suggests evaluating performance on the full dataset to assess scalability. While this is a valid consideration, the core design philosophy of GeoPurify is to achieve robust 3D understanding with _minimal_ data under resource-constrained conditions, rather than relying on massive-scale training. To empirically address the concern regarding data sensitivity and scalability, we conducted an ablation study expanding the training set to include 10, 20, 30, and 50 scenes.
>
> | Training Subset Size | mIoU (%) | mAcc (%) |
> | :------------------: | :------: | :------: |
> |      10 scenes       |   54.7   |   72.4   |
> |    **20 scenes**     | **55.1** | **72.5** |
> |      30 scenes       |   55.1   |   72.5   |
> |      50 scenes       |   55.0   |   72.5   |
>
> As shown in the table above, increasing the data volume from 10 to 20 scenes yields a performance boost. However, beyond this range (30 and 50 scenes), further increases yield diminishing returns, with metrics stabilizing. **This trend validates the hypothesis that because geometric structures (e.g., surface continuity, adjacency) are repetitive and universal, the Student Affinity Network captures the necessary class-agnostic geometric priors efficiently from a small subset.** This plateau indicates that the framework effectively maximizes the potential of the current VLM features. Consequently, the method’s performance is correlated with the semantic quality of the VLM backbone. As VLM architectures evolve, this data-efficient framework can be applied to advanced models to achieve sustainable performance gains without dependence on scaling up 3D data annotation.
>
> > Q3:Is the point cloud required as input at inference time? From the text it seems to be the case (line 155), but from Figure 2 it seems to be needed only at training time.
>
> We apologize for the initial submission's unclear wording, which led to misunderstandings. We clarify that the inference stage does not necessitate an external or original raw point cloud as input. Instead, the system operates by aggregating the projections of the multi-view images into a point cloud representation. The Student Affinity Network then processes these aggregated coordinates to generate the geometric embeddings required for the pooling mechanism. The specific path involving the "3D Geometric SSL" Teacher model, as shown in Figure 2, is strictly for the training phase. We have provided an explanation in Section 3.2.

---

> ### Author Response · Authors · 2025-11-27
>
> Dear Reviewer tyeq,
>
> Thank you once again for your insightful and constructive review, which has greatly contributed to improving the quality and clarity of our paper Geopurify. We sincerely hope that our rebuttal has satisfactorily addressed your questions and concerns.
>
> If you have any remaining questions or would like further clarification, we would be happy to provide it. We truly appreciate your valuable feedback and the time and effort you dedicated to reviewing our work.
>
> Best regards,
>
> Authors of Submission 1462

---

### Official Review · Reviewer_La46 · 2025-11-01

**Soundness:** 3
**Presentation:** 3
**Contribution:** 1
**Rating:** 2
**Confidence:** 4

**Summary:**

This paper presents GeoPurify, a geometric distillation framework for open-vocabulary 3D semantic segmentation. By introducing geometry-guided contrastive distillation and affinity-based semantic purification, the method aims to reduce projection noise from 2D VLM features and improve 3D semantic consistency. The approach demonstrates strong results on datasets and requires only 1.5 % of the trainning data, showing satisfactory annotation efficiency.

**Strengths:**

1. The idea of leveraging geometric priors to purify VLM features is meaningful and addresses a real weakness of projection-based pipelines.

2. The method achieves competitive performance with significantly reduced 3D training data, which is practically valuable for indoor scene understanding.

**Weaknesses:**

1. The claimed paradigm shift from *“Segmentation-and-Matching”* to *“Segmentation-as-Understanding”* is not entirely novel, as recent works such as OV3D[1] and PGOV3D[2] already adopt VLM-based understanding pipelines. The conceptual contribution could be more precisely positioned, and comparisons with these recent methods are somewhat limited.

[1] Jiang, Li, Shaoshuai Shi, and Bernt Schiele. "Open-vocabulary 3d semantic segmentation with foundation models" in Proceedings of the IEEE/CVF Conference on Computer Vision and Pattern Recognition, 2024.

[2] Zhang, Shiqi, et al. "PGOV3D: Open-Vocabulary 3D Semantic Segmentation with Partial-to-Global Curriculum" in Proceedings of ACM Multimedia, 2025.

2. The current subset selection depends on global dataset statistics, which contradicts the “fully annotation-free” motivation. It remains unclear whether the reported gains are due to the method itself or a bias toward selecting higher-quality samples.

3. The method is evaluated only on indoor datasets (e.g., ScanNet). Since the approach relies heavily on geometric consistency, its generalization to outdoor datasets such as nuScenes remains unclear.

**Questions:**

1. Could the authors provide a quantitative comparison with recent VLM-based understanding pipelines?

2. How robust is GeoPurify if the subset is selected randomly instead of using global statistics?

3. How does the method perform on outdoor datasets such as nuScenes or SemanticKITTI?

---

> ### Author Response · Authors · 2025-11-21
>
> We appreciate the reviewer La46 recognition of the value of leveraging geometric priors to purify VLM features and our method's competitive performance under low-data regimes. We address the concerns regarding our contribution positioning, subset selection, and generalization below.
>
> > W1:The claimed paradigm shift from “Segmentation-and-Matching” to “Segmentation-as-Understanding” is not entirely novel, ...The conceptual contribution could be more precisely positioned, and comparisons with these recent methods are somewhat limited.
>
> We acknowledge that the concept of leveraging VLM-based understanding pipelines has appeared in recent literature, and we have revised our manuscript to more precisely articulate our specific technical contribution. Our novelty centers on the hypothesis that **VLM-projected features not only contain rich semantic information but also embed a latent 3D geometric structure.** Unlike prior methods that often attempt to learn geometry and semantics simultaneously—a data-intensive process—GeoPurify demonstrates that recovering this latent structure via a student affinity network is a significantly more efficient path to 3D segmentation. Specifically, our framework leverages a generalist VLM to establish a high semantic ceiling and introduces a "Geometric Contrastive Distillation" mechanism to distill robust structural affinities from unlabeled 3D scans. This approach effectively decouples the semantic richness of VLMs from the geometric consistency of 3D teachers. In comparison, while methods like OV3D and PGOV3D also utilize VLM capabilities, they typically rely on large-scale training data to align features or learn curriculum strategies, making them less effective in data-constrained scenarios. In contrast, GeoPurify achieves competitive performance using only ~1.5% of the training data by focusing on the _purification_ of these dense geometric affinities. We have updated our Introduction and Related Work section to explicitly discuss these distinctions, highlighting our unique advantage in data-efficient settings.
>
> > Q1:Could the authors provide a quantitative comparison with recent VLM-based understanding pipelines?
>
> **Table 1: Quantitative results for open-vocabulary 3D semantic segmentation on ScanNetV2 and Matterport3D**
>
> | Type               | Method               | 3D Net Training Data | ScanNetV2 mIoU | ScanNetV2 mAcc | Matterport3D mIoU | Matterport3D mAcc |
> | :----------------- | :------------------- | :------------------: | :------------: | :------------: | :---------------: | :---------------: |
> | **Zero-shot**      | DMA-text only        |         100%         |      50.5      |      63.7      |       39.8        |       49.5        |
> |                    | DMA-3D               |         100%         |      54.8      |      66.9      |         -         |         -         |
> |                    | OpenScene-3D         |         100%         |      51.6      |      63.1      |       40.5        |       48.8        |
> |                    | CUA-O3D (3D)         |         100%         |      54.1      |      64.1      |       41.3        |       49.5        |
> |                    | GGSD                 |         100%         |      56.5      |      68.6      |         -         |         -         |
> |                    | **OV3D**             |       **100%**       |    **57.3**    |    **72.9**    |     **45.8**      |     **62.4**      |
> |                    | **PGOV3D**           |       **100%**       |    **59.5**    |    **73.2**    |         -         |         -         |
> | **Data-Efficient** | CUA-O3D (reimple)    |        ~1.5%         |      18.1      |      26.4      |       14.0        |       20.5        |
> |                    | **(Ours) GeoPurify** |      **~1.5%**       |    **55.1**    |    **72.5**    |     **40.2**      |     **62.4**      |
>
> We have updated **Table 1** to include the reported performance of OV3D (CVPR 2024) and PGOV3D (ACM MM 2025). OV3D reports an mIoU of 57.3 and PGOV3D reports 59.5 on ScanNetV2. While these baselines achieve strong performance, GeoPurify (55.1 mIoU) remains highly competitive, particularly given the immense disparity in training resources. It is crucial to clarify that we could not evaluate these methods on our specific 1.5% data subset as their code has not been made public. However, comparisons must account for data scale: both OV3D and PGOV3D rely on training with **100%** of the available dataset to achieve their reported metrics. In stark contrast, GeoPurify utilizes only **1.5%** of **unlabeled** 3D scans to reach a comparable performance level (e.g., within ~2 mIoU of OV3D). This underscores our core contribution: GeoPurify demonstrates that competitive 3D segmentation can be achieved via _geometric purification_ of latent features, offering a significantly more data-efficient path than massive supervised scaling. We have updated the Related Work section to discuss these distinctions in detail.

---

> ### Author Response · Authors · 2025-11-21
>
> > W2 & Q2: The current subset selection depends on global dataset statistics, which contradicts the “fully annotation-free” motivation. It remains unclear whether the reported gains are due to the method itself or a bias toward selecting higher-quality samples. How robust is GeoPurify if the subset is selected randomly instead of using global statistics?
>
> We address the concern regarding the dependency on global dataset statistics by evaluating GeoPurify using a purely random subset selection strategy. As shown in Table R1 below, the model trained on randomly selected scenes achieves an mIoU of 54.64 and mAcc of 72.16. This performance is statistically comparable to our heuristic-based selection (55.1 mIoU / 72.5 mAcc), with a negligible drop of roughly 0.5 mIoU. This empirical evidence confirms that **the reported gains are driven by the efficacy of the geometric purification method itself, not by a bias toward high-quality samples.** The heuristic selection strategy is employed simply to ensure a broader coverage of diverse environmental archetypes (e.g., ensuring both bathrooms and offices are represented) when the sample size is extremely small, but the method remains robust without it. Furthermore, to better align with the "fully annotation-free" paradigm while preserving data diversity, we explored a "Coordinate and Color Clustering" strategy that selects representative scenes based solely on unsupervised clustering of raw RGB values and spatial coordinates. This method achieves 54.9 mIoU and 72.3 mAcc, yielding results nearly identical to ours heuristic approach. **Collectively, these experiments demonstrate that our performance improvements are intrinsic to the proposed geometric purification mechanism and do not stem from a reliance on global statistical information for data selection.** We have included the detailed analysis in the Appendix.
>
> **Table R1: Robustness Analysis of Subset Selection Strategy (ScanNetV2)**
> | Selection Strategy | Training Data | mIoU | mAcc |
> | :--- | :--- | :--- | :--- |
> | Heuristic (Ours) | ~1.5% (20 scenes) | 55.1 | 72.5 |
> | Random | ~1.5% (20 scenes) | 54.6 | 72.2 |
> |Coordinate and Color Clustering|~1.5% (20 scenes)|54.9|72.3|
>
> > W3 & Q3: The method is evaluated only on indoor datasets (e.g., ScanNet). Since the approach relies heavily on geometric consistency, its generalization to outdoor datasets such as nuScenes remains unclear. How does the method perform on outdoor datasets such as nuScenes or SemanticKITTI?
>
> We recognize the importance of evaluating generalization in outdoor environments like nuScenes. However, conducting native training for GeoPurify on such datasets presents a specific resource constraint: while codebases for 3D self-supervised learning exist (Sonata [1], PTv3 [2], MSC [3]), publicly available pre-trained weights are predominantly limited to indoor scenarios. The lack of released pre-trained weights for outdoor domains prevented us from distilling a domain-specific geometric teacher without computationally expensive pre-training from scratch. To address the reviewer's concern within these constraints, we designed an experiment to **evaluate the zero-shot generalization of our Geopurify framework, trained on the indoor dataset (ScanNet), when directly applied to the outdoor dataset (nuScenes)**. Following the standard evaluation protocol and label mapping (`map_nuscenes_details`) from OpenScene, first, we established a baseline using the 2D VLM without the geometry-guided pooling mechanism on the full nuScenes validation set. The method achieves an mIoU of 30.22 and an mAcc of 47.17. Second, upon applying GeoPurify, which integrates the 2D VLM with our geometry-guided pooling mechanism trained solely on ScanNet, performance improved to 31.57 mIoU and 47.81 mAcc. Compared to the baseline using only 2D VLM features, this improvement (+1.35 mIoU) demonstrates that **the geometric affinities distilled from indoor environments possess a degree of generalization, successfully enforcing latent structural consistencies even across sensor domains.**
>
> [1]Wu, Xiaoyang, et al. "Sonata: Self-supervised learning of reliable point representations." Proceedings of the Computer Vision and Pattern Recognition Conference. 2025.
>
> [2]Wu, Xiaoyang, et al. "Point transformer v3: Simpler faster stronger." Proceedings of the IEEE/CVF conference on computer vision and pattern recognition. 2024.
>
> [3]Wu, Xiaoyang, et al. "Masked scene contrast: A scalable framework for unsupervised 3d representation learning." Proceedings of the IEEE/CVF Conference on computer vision and pattern recognition. 2023.

---

> ### Author Response · Authors · 2025-11-27
>
> Dear Reviewer La46,
>
> Thank you once again for your insightful and constructive review, which has greatly contributed to improving the quality and clarity of our paper Geopurify. We sincerely hope that our rebuttal has satisfactorily addressed your questions and concerns.
>
> If you have any remaining questions or would like further clarification, we would be happy to provide it. We truly appreciate your valuable feedback and the time and effort you dedicated to reviewing our work.
>
> Best regards,
>
> Authors of Submission 1462

---

### Author Response · Authors · 2025-12-01

Dear PCs, SACs, ACs, and all of our reviewers,

We sincerely thank you for your thorough reviews and valuable feedback. Your comments and discussions have greatly helped us in improving our work, and we have answered and addressed the concerns raised.

We appreciate your recognition of our paper's strengths:

- Reviewers La46 and MQX3 found the idea of leveraging geometric priors **"meaningful and addresses a real weakness of projection-based pipelines"**, representing a **"principled shift in perspective toward 'segmentation as understanding' "** where recovering latent structure is **"novel and well-motivated"**.
- Reviewers La46 and 1k2F highlighted that our method achieves competitive results while **"significantly reducing reliance on costly 3D annotations"**, offering a **"practically valuable"** solution for indoor scene understanding.
- Reviewers tyeq and 1k2F praised the cross-modal distillation strategy as **"conceptually sound and well motivated"**, noting that it **"addresses the key issue of geometric fragmentation"** resulting from 2D-to-3D projection.

Reviewer La46 comments that our stated paradigm shift from "Segmentation-and-Matching" to "Segmentation-as-Understanding" is similar to recent VLM-based works (e.g., OV3D, PGOV3D), and suggests more precisely positioning our contributions. We have **reorganized the Introduction and Related Work sections** to explicitly articulate our unique contribution. We emphasize that unlike prior methods that rely on massive supervision to simultaneously learn geometry and semantics, our approach centers on the VLM features already embed **latent structure** that can be efficiently recovered via geometric distillation, **rather than achieving the paradigm shift** from "Segmentation-and-Matching" to "Segmentation-as-Understanding". This center allows for extreme data efficiency. To empirically validate this, we presented a comparative analysis showing that GeoPurify achieves competitive performance (55.1 mIoU) using only **~1.5% of unlabeled training data**, whereas competitors like OV3D require **100% of the dataset** to reach similar levels (57.3 mIoU), effectively highlighting our advantage in data-efficient utilization.

We have also addressed the other primary concerns raised during the discussion period:

- Regarding the concern that our **heuristic subset selection might introduce a bias toward high-quality samples** (Reviewers La46), we conducted an ablation studies using purely random selection and "Coordinate and Color Clustering" both showed negligible performance drops (0.2–0.5 mIoU) compared to the heuristic strategy, confirming that our gains stem from the geometric purification method itself rather than a bias in data selection.
- In response to questions regarding **scalability and the sensitivity of our method to training data size** (Reviewers tyeq, 1k2F), we performed an ablation study with 10, 20, 30, and 50 scenes. The observed performance plateau at 20 scenes validates our hypothesis that class-agnostic geometric affinities are repetitive and universal, allowing for efficient convergence on small datasets without needing large-scale data.
- Addressing the concern that our **geometry-reliant method might not generalize to outdoor domains like nuScenes** (Reviewer La46), we use zero-shot setting to evaluated GeoPurify on the outdoor nuScenes dataset. Despite being trained solely on indoor ScanNet data, our framework improved performance by +1.35 mIoU over the baseline, demonstrating that the distilled geometric affinities possess cross-domain robustness.
- Regarding the request for **comparisons with recent VLM pipelines and instance segmentation methods** (Reviewers La46, 1k2F), we updated our comparisons to include OV3D and PGOV3D and clarified the task distinction regarding instance segmentation methods like Open3DIS. Additionally, to prove our method is **independent of the specific VLM backbone**, we validated our framework on a standard LSeg backbone, achieving consistent gains (+2.6 mIoU).

Finally, we thank the reviewers for their constructive comments. We believe these additional results and clarifications firmly establish GeoPurify as a data-efficient, robust solution for open-vocabulary 3D segmentation.

Thank you again for your time and efforts.

Best regards,

The Authors of Submission 1462

---

### Meta-Review · Area_Chair_kmvD · 2025-12-31

**Summary:**

The reviewers generally appreciate the contributions of the paper, particularly in advancing 3D perception with improved data efficiency. The primary concerns raised across reviews relate to: comparisons with recent methods, justification for subset selection strategies, the impact of data scale on performance, reliance on pre-trained models, and evaluation on diverse datasets. The authors have addressed most of these concerns through additional experiments, analyses, and discussions. One lingering issue is the limited performance gain over zero-shot baselines, which remains partially unresolved. Considering the overall strength of the paper and the substantial rebuttal, I recommend acceptance.

**Reviewer Concerns:**

Most of the reviewers' concerns were effectively addressed by the authors. The paper now includes detailed comparisons with recent methods such as OV3D and PGOV3D, and it also expands the discussion to include Open3DIS and UniSeg3D, which were initially identified as gaps. Additionally, the concerns regarding subset selection and the choice of training scenes have been resolved through further ablation studies, while performance on outdoor datasets has been explored to some extent. However, Reviewer tyeq raised a concern about the limited performance gain over zero-shot baselines, which the authors partially addressed but did not fully resolve. Despite conducting additional experiments, the observed improvements remain modest, which somewhat weakens the empirical claims. Nonetheless, the authors have made meaningful contributions to the field, particularly in terms of data efficiency, and all other concerns have been sufficiently addressed.

**Reviewer Scores:**

In terms of post-rebuttal scores, Reviewer La46, who initially rated the paper a 2, is likely to raise his/her score to a positive score, given that all his/her concerns were addressed. Reviewer tyeq, with an original score of 6, is expected to remain cautious, given that the limited performance gain over zero-shot baselines was only partially resolved. Reviewer MQX3, who gave the paper an 8, will likely maintain high score, as his/her concern about pre-trained models was adequately addressed. Finally, Reviewer 1k2F, initially scoring 6, is likely to maintain the positive score or slightly increase score to 8, given that all his/her concerns were resolved, though with lower confidence in the rebuttal.  Overall, AC believes that the majority of reviewers will support acceptance after considering the rebuttal.

---

### Decision · Program_Chairs · 2026-01-26

Accept (Poster)